# MODEL SHAPLEY: Find Your Ideal Parameter Player via One Gradient Backpropagation

**Xu Chu**[*] [1,2,4]
chu_xu@pku.edu.cn

**Xinke Jiang**[1,3,4]
xinkejiang@stu.pku.edu.cn

**Rihong Qiu**[1,3,4]
rihongqiu@stu.pku.edu.cn

**Jiaran Gao**[1,3,4]
jiarangao@stu.pku.edu.cn

**Junfeng Zhao**[1,4,5]
zhaojf@pku.edu.cn

[1]Key Laboratory of High Confidence Software Technologies, Ministry of Education, Beijing, China
[2]Center on Frontiers of Computing Studies, Peking University, Beijing, China
[3]National Engineering Research Center For Software Engineering, Peking University, Beijing, China
[4]School of Computer Science, Peking University, Beijing, China
[5]Big Data Technology Research Center, Nanhu Laboratory, Jiaxing, China

https://github.com/Artessay/ModelShapley

## Abstract

Measuring parameter importance is crucial for understanding and optimizing large language models (LLMs). Existing work predominantly focuses on pruning or probing at neuron/feature levels without fully considering the cooperative behaviors of model parameters. In this paper, we introduce a novel approach–MODEL SHAPLEY to quantify parameter importance based on the Shapley value, a principled method from cooperative game theory that captures both individual and synergistic contributions among parameters, via only one gradient backpropagation. We derive a scalable second-order approximation to compute Shapley values at the parameter level, leveraging blockwise Fisher information for tractability in large-scale settings. Our method enables fine-grained differentiation of parameter importance, facilitating targeted knowledge injection and model compression. Through mini-batch Monte Carlo updates and efficient approximation of the Hessian structure, we achieve robust Shapley-based attribution with only modest computational overhead. Experimental results indicate that this cooperative game perspective enhances interpretability, guides more effective parameter-specific fine-tuning and model compressing, and paves the way for continuous model improvement in various downstream tasks.

## 1 Introduction

Many of today's most powerful deep neural networks contain hundreds of millions or even billions of parameters, but *not all parameters are created equal*. From the perspective of **parallel distributed processing** (PDP) [1], the emergent behavior of a network arises from collective interactions among numerous parameters, each of which may appear individually unimportant but becomes crucial in synergy with others. Identifying precisely which weights drive performance is therefore essential for tasks such as **knowledge injection** [2], **interpretability** [3], and **model compression** [4]. While heuristic approaches like weight magnitude [5] or first-order sensitivity [6] provide rough estimates of parameter importance, they often neglect the synergy among parameters—i.e., a weight might appear "unimportant" on its own yet become critical when combined with certain others [6].

---

[*]Corresponding author.

39th Conference on Neural Information Processing Systems (NeurIPS 2025).

To address this shortcoming, we introduce **Model Shapley**: a cooperative game-theoretic framework for quantifying each parameter's contribution in a way that naturally accounts for *interactions*. Specifically, we treat each parameter as a "player" in a game and measure its **Shapley value** [7]—the average marginal improvement it delivers over all possible subsets of other parameters. The Shapley value is known for satisfying desirable fairness axioms, but computing it exactly requires evaluating an exponential number of parameter subsets [8]. This has historically limited its use for parameter attribution.

**MODEL SHAPLEY** overcomes this barrier through a **second-order, closed-form approximation** that exploits local curvature of the loss (via the Hessian or Fisher information). With a single forward–backward pass and a blockwise Fisher-based Hessian approximation [9], we estimate every parameter's Shapley value at scale. Further, we introduce a **Monte Carlo incremental** scheme that updates these importance scores online, accommodating large datasets or streaming scenarios without repeatedly retraining or pruning.

We demonstrate in extensive experiments that **MODEL SHAPLEY** *significantly outperforms* standard heuristics in three key applications:

- **Knowledge Injection.** By protecting or regularizing high-Shapley parameters, we reduce catastrophic forgetting [2], since these weights carry more critical knowledge from previous tasks.
- **Interpretability.** Our framework supplies *parameter-level* attribution, revealing how specific weights or neurons most heavily influence the network's outputs, complementing traditional input-based explanations [3].
- **Quantization/Compression.** Shapley-aware importance scores guide pruning or mixed-precision quantization [4] more effectively than magnitude or naive second-order methods, preserving accuracy even under aggressive compression.

**Contributions.** (1) We formulate the parameter-importance problem using the Shapley value, offering a principled way to capture cross-parameter interactions—aligning with the PDP view that cooperative effects among parameters are central to network function. (2) We derive a *second-order, closed-form* approximation (and an incremental extension) for computing these values in a single pass, scalable to modern architectures. (3) We thoroughly validate **MODEL SHAPLEY** on knowledge injection, interpretability, and compression, demonstrating its superior performance compared to simpler heuristics. Overall, this game-theoretic lens provides a fresh perspective on how to *analyze* and *optimize* deep neural networks at the parameter level.

## 2 Related Work

### 2.1 Shapley Value

***The Shapley value (SV)***, initially introduced by Shapley [7], provides a principled framework for fair payoff allocation among players in cooperative games. Building upon this theoretical foundation, SV has been adopted across a variety of domains over the decades. [10] extended SV into probabilistic settings, while [11] explored interaction indices among players. In data-driven fields, SV has increasingly been leveraged for model interpretability and data valuation. [3] popularized feature-level explanations via SHAP, [12] introduced Data Shapley to quantify the contribution of individual training examples. To address the exponential complexity of SV computation, several works have proposed approximation algorithms [13, 8], i.e., [14] approximate data SV in one training run. Recent comprehensive surveys, such as [15], consolidate SV's applications throughout the entire data analytics lifecycle, identifying major challenges related to efficiency, approximation, privacy, and interpretability. Beyond data and feature levels, SV has also been applied at the model level. Some works utilize Shapley value to explain the information flow mechanisms inside neural networks [13, 16], while others investigate model-level interactions, for example in federated learning [17] and in traditional machine learning models such as random forests [18, 19]. However, despite these advances, existing studies predominantly focus on applying SV at the data sample or feature level, or at the model level for inter-model comparisons. Due to the intrinsic combinatorial explosion, **only few prior work has attempted to extend Shapley Value to the level of neural network internals [20, 21]**, aiming to compute cooperative game values for neurons. Nevertheless, these works rely on a multi-armed bandit and Monte Carlo approximation with significant computational overhead, and the random permutation of neurons can harm model performance, introducing estimation bias. This gap motivates our exploration of neuron-level Shapley analytics.

## 2.2 Parameter Importance Quantification in Deep Learning

In recent years, with the growing interest in understanding the internal mechanisms of neuron networks, ***Parameter Importance Quantification*** [22, 23, 5] has become a crucial research direction. Analogous to the functional partitioning observed in the human cortex [24, 25, 26, 27]—where different regions are responsible for different cognitive tasks—LLMs may also develop "functional partitions" during training, with certain structural units (e.g., attention heads [28, 29, 30], feed-forward networks (FFNs) [31, 32, 33], individual neurons [34, 35, 36, 37, 5, 38]) specializing in specific tasks or storing particular knowledge. Parameter importance quantification typically involves clarifying the target capabilities (e.g., robustness [35], knowledge storage [29, 34]), preparing appropriate probing datasets, and determining the granularity of analysis (e.g., attention head [39, 29], neuron [37, 40], or layer [31]). Existing localization methods can be broadly categorized into two groups: ❶ methods with forward signals, which locate functional units based on activation statistics (e.g., LAPE [37], Locate-then-unlearn [40]), activation patching (e.g., Function Vector [39], FVG [41]), or integrated gradients [42]; ❷ and methods involving backward signals, which either trace gradient trajectories (e.g., Recurrent-KIF [43], TaSL [35], OBS [5], OBD [38], GPTQ [4], DFAMS [44], LightGNTK [45]) or combine forward and backward propagation signals (e.g., Parenting [36]) to dynamically discover functional modules. **However, these works only consider the intrinsic importance of parameter units for specific tasks, while the utility of the huge cooperative game between parameters is ignored**.

There is a conceptually related framework dubbed Component Attribution via Regression (COAR) [46]. We would like to highlight the differences: While both **MODEL SHAPLEY** and COAR tackle the challenge of parameter synergy, they are motivated by different goals, parameter importance quantification for **MODEL SHAPLEY** versus model editing for COAR. Besides, two frameworks employ different technical means. COAR learns a surrogate linear model, capturing interactions without second-order curvature. Whereas **MODEL SHAPLEY** directly derives a closed-form approximation of the Shapley value from the model's loss landscape.

# 3 Background of Shapley

The Shapley value [7, 47], originally developed in cooperative game theory [48], provides a principled approach to quantifying the marginal contribution of individual elements within a set. In this section, we introduce the notion of a utility function and formally define the Shapley value.

***Definition 1. (Utility Function)*** Let $\Theta := \{\theta_i\}_{i=1}^M$ be a finite set, where each element $\theta_i$—such as a data point, a feature attribution, or a model parameter—is regarded as a ***player***. A utility function $\mathbf{U} : 2^\Theta \to \mathbb{R}$ assigns a real-valued score to each subset $\Theta_\mathcal{S} \subseteq \Theta$, representing the utility or value of that subset within the context of a given learning task.

***Definition 2. (Shapley Value)*** The Shapley value $\phi_\theta(\mathbf{U})$ for an element $\theta \in \Theta$ quantifies its average marginal contribution to the utility function, computed over all possible subsets of $\Theta$ that exclude $\theta$. It is defined as:

$$\phi_\theta(\mathbf{U}) := \frac{1}{M} \sum_{k=1}^M \binom{M-1}{k-1}^{-1} \sum_{\substack{\Theta_\mathcal{S} \subseteq \Theta \setminus \{\theta\} \\ |\Theta_\mathcal{S}| = k-1}} [\mathbf{U}(\Theta_\mathcal{S} \cup \{\theta\}) - \mathbf{U}(\Theta_\mathcal{S})] \tag{1}$$

where $M = |\Theta|$ denotes the total number of elements. This value satisfies four desirable axioms that ensure fairness in attribution: ❶ the *Null Player* property, stating that if an element $\theta$ has no effect on any subset, i.e., $\mathbf{U}(\Theta_\mathcal{S} \cup \{\theta\}) = \mathbf{U}(\Theta_\mathcal{S})$ for all $\Theta_\mathcal{S}$, then $\phi_\theta(\mathbf{U}) = 0$; ❷ the *Symmetry* property, which guarantees that if two elements contribute identically to all subsets, they receive equal value, i.e., $\phi_{\theta_i}(\mathbf{U}) = \phi_{\theta_j}(\mathbf{U})$; ❸ the *Linearity* property, stating that the value respects linear combinations of utility functions, so that for any scalars $\alpha_1, \alpha_2 \in \mathbb{R}$, it holds that $\phi_\theta(\alpha_1 U_1 + \alpha_2 U_2) = \alpha_1 \phi_\theta(U_1) + \alpha_2 \phi_\theta(U_2)$; and finally, ❹ the *Efficiency* property, which ensures that the total utility is fully distributed across all elements, i.e., $\sum_{\theta \in \Theta} \phi_\theta(\mathbf{U}) = \mathbf{U}(\Theta)$.

# 4 MODEL SHAPLEY

In this section, we introduce **MODEL SHAPLEY**, a scalable framework for estimating Shapley values at the parameter level, which consists of the following three key components:

- **Section 4.1: Closed-Form Shapley Derivation.** We theoretically derive a closed-form expression for the parameter-wise Shapley value, showing that it can be efficiently computed using a single backward pass through the network, thereby eliminating the need for repeated marginal evaluations.

- **Section 4.2: Monte Carlo-Based Shapley Incremental Approximation.** To address real-world scenarios where the full dataset may exceed memory capacity (e.g., streaming scenarios or online learning), we develop an incremental Monte Carlo approximation scheme. This method provides online estimates of Shapley values through mini-batch updates, implemented via an exponentially weighted moving average (EWMA).

- **Section 4.3: Fisher Approximation with Blockwise Regularization.** To reduce the computational cost associated with computing the full Hessian matrix, we approximate it using the empirical Fisher Information Matrix. Furthermore, we apply a blockwise regularization strategy to improve stability and efficiency during estimation.

The main notations, algorithm, and the application of **MODEL SHAPLEY** are listed in Appendix B, Appendix G, and Appendix E, respectively.

## 4.1 Closed-Form Shapley Derivation

Computing the Shapley value of individual parameter units (especially neurons) in neural networks—particularly in large-scale architectures such as LLMs or VLMs—is computationally prohibitive. Formally, evaluating the marginal utility term $\mathbf{U}(\Theta_S \cup \{\theta\}) - \mathbf{U}(\Theta_S)$ for all subsets $\Theta_S \subseteq \Theta \setminus \{\theta\}$ requires enumerating an exponential number of combinations. For each subset, this process involves removing a specific parameter $\theta_i$ and re-evaluating (or even re-training) the model to assess the change in performance on a held-out test set. In large models with hundreds of millions or billions of parameters (e.g., 0.5B, 7B, or even 670B in LLMs), the number of subsets grows exponentially, yielding a worst-case complexity of $\mathcal{O}(2^M)$, where $M$ is the total number of model parameters. Even with substantial computational resources, the sheer number of forward and backward passes—plus potential fine-tuning—renders exact Shapley value computation infeasible in practice.

However, we observe that the structure of trained neural networks enables a tractable approximation. Specifically, we can estimate the marginal utility of an individual parameter through the change in training loss when that parameter is **removed**—i.e., deactivated by setting it to zero or frozen during evaluation. Instead of relying on a local Maclaurin expansion at the origin [49]—which may yield inaccurate approximations when the parameter value is far from zero—we adopt a path-integrated formulation [50]. In this view, the **loss change induced by parameter removal is modeled as a continuous trajectory in parameter space**, and the total effect is computed by integrating the gradient along this path.

**Theorem 4.1** (Path-Integrated Approximation of the Loss). *Let $\Theta^\tau = \{\theta_1^\tau, ..., \theta_M^\tau\}$ be the current parameter vector at iteration $\tau$. For any perturbed configuration $\Theta'$ defined along a linear path from $\Theta^\tau$, the loss function $\mathcal{L}(\Theta'; x, y)$ evaluated on an input–label pair $(x, y)$ admits the following integral approximation:*

$$\mathcal{L}(\Theta'; x, y) = \mathcal{L}(\Theta^\tau; x, y) + \int_0^1 \nabla_\Theta \mathcal{L}(\Theta_t; x, y)^\top \frac{d\Theta_t}{dt} dt, \tag{2}$$

*where $\Theta_t := \Theta^\tau + t(\Theta' - \Theta^\tau)$ defines the linear interpolation between $\Theta^\tau$ and $\Theta'$.*

❶ **Leave-One-Parameter-Out Analysis** We now quantify the loss increase caused by removing a single parameter unit. Let $\theta_i^\tau$ be the $i$-th parameter at step $\tau$, and let $e_i$ be the standard basis vector with 1 at position $i$. Define the perturbed parameter vector as $\Theta^{\tau, -i} := \Theta^\tau - \theta_i^\tau e_i$, which corresponds to deactivating parameter $\theta_i$ while keeping all others fixed. To estimate the associated loss change, we adopt a path-integrated formulation by defining a linear interpolation between the removed and original states: $\Theta_t := \Theta^{\tau, -i} + t \cdot \theta_i^\tau e_i, t \in [0, 1]$, and interpret the loss difference as the cumulative gradient along this path: $\Delta\mathcal{L}(\theta_i^\tau) := \mathcal{L}(\Theta^{\tau, -i}; x, y) - \mathcal{L}(\Theta^\tau; x, y) = -\int_0^1 \frac{\partial \mathcal{L}(\Theta_t)}{\partial \theta_i} \cdot \theta_i^\tau dt$. This yields the following approximation as (Appendix C.1 for details):

**Theorem 4.2** (Loss Change from Single Parameter Removal). *Let $\theta_i^\tau$ be the value of parameter $i$ at step $\tau$. Then the loss increase after removing $\theta_i$ is approximately:*

$$\Delta\mathcal{L}(\theta_i^\tau) := \mathcal{L}(\Theta^{\tau,-i};x,y) - \mathcal{L}(\Theta^\tau;x,y) \approx -\mathbf{g}_i^\tau\theta_i^\tau + \tfrac{1}{2}\mathbf{w}_{ii}^{(i)}\mathbf{H}_{ii}^\tau(\theta_i^\tau)^2, \tag{3}$$

*where $\mathbf{g}_i^\tau = \frac{\partial\mathcal{L}(\Theta^\tau)}{\partial\theta_i}$ is the gradient of loss $\mathcal{L}(\Theta^\tau)$ with respect to parameter $\theta_i$ at iteration $\tau$, $\mathbf{H}_{ii}^\tau$ is the $i$-th diagonal entry of the Hessian, and $\mathbf{w}_{ii}^{(i)} \in [0,1]$ is a curvature-based path weight that averages second-order effects along the trajectory.*

*Remark* 4.3 (From Loss Change to Marginal Utility). This loss change conceptually aligns with our goal of measuring the impact of removing a parameter unit. **Although model performance is typically assessed through discrete metrics (e.g., test accuracy), in differentiable settings, such performance shifts can be naturally approximated via continuous changes in the loss.** Thus, defining the utility function as $\mathbf{U}(\Theta) := -\mathcal{L}(\Theta;x,y)$, the marginal utility of parameter $\theta_i^\tau$ becomes: $\mathbf{U}(\Theta^\tau) - \mathbf{U}(\Theta^{\tau,-i}) = \mathcal{L}(\Theta^{\tau,-i};x,y) - \mathcal{L}(\Theta^\tau;x,y) \approx -\mathbf{g}_i^\tau\theta_i^\tau + \tfrac{1}{2}\mathbf{w}_{ii}^{(i)}\mathbf{H}_{ii}^\tau(\theta_i^\tau)^2$. Notably, the first-order term $|g_i^\tau\theta_i^\tau|$ recovers importance scores used in prior work on pruning and saliency [51, 49]. However, relying solely on this term ignores interactions between parameters. These interactions are partially captured by second-order terms, particularly in overparameterized networks such as large language models [52].

**❷ Approximation of Marginal Utility**  While removing a single parameter offers useful insights, it fails to capture cooperative interactions among parameters—critical for Shapley-style attribution. Therefore, we extend our analysis to arbitrary subsets. Let $\mathcal{S} \subseteq \{1,...,M\}$ be a subset of parameter indices, and define $\Theta^{\tau,-\mathcal{S}} := \Theta^\tau - \sum_{i\in\mathcal{S}}\theta_i^\tau e_i$, which corresponds to setting all parameters in $\mathcal{S}$ to zero. To estimate the corresponding loss increase, we adopt a path-integrated view over the subspace of removed parameters. Thus we have (Appendix C.2 for details):

**Theorem 4.4** (Loss Change from Removing a Parameter Subset). *Let $\mathcal{S} \subseteq \{1,...,M\}$ be a subset of parameter indices. Then the loss increase from removing parameters in $\mathcal{S}$ is approximated as:*

$$\Delta\mathcal{L}(\Theta^{\mathcal{S}}) := \mathcal{L}(\Theta^{\tau,-\mathcal{S}};x,y) - \mathcal{L}(\Theta^\tau;x,y) \approx -\sum_{i\in\mathcal{S}}\mathbf{g}_i^\tau\theta_i^\tau + \tfrac{1}{2}\sum_{i,j\in\mathcal{S}}\mathbf{w}_{ij}^{(\mathcal{S})}\mathbf{H}_{ij}^\tau\theta_i^\tau\theta_j^\tau, \tag{4}$$

*where $\mathbf{g}_i^\tau = \frac{\partial\mathcal{L}(\Theta^\tau)}{\partial\theta_i}$ is the gradient of loss $\mathcal{L}(\Theta^\tau)$ with respect to parameter $\theta_i$ at iteration $\tau$, $\mathbf{H}_{ij}^\tau$ is the $(i,j)$-th entry of the Hessian matrix, and $\mathbf{w}_{ij}^{(\mathcal{S})} \in [0,1]$ denotes a curvature-aware path weight reflecting the average influence of parameters $i$ and $j$ along the joint removal path.*

*Remark* 4.5 (Efficient Approximation of Marginal Contributions). This generalization enables efficient approximation of Shapley-style marginal utility terms. Given a coalition $\mathcal{S} \subseteq \Theta \setminus \{i\}$, the marginal contribution of parameter $\theta_i$ is: $\mathbf{U}(\Theta^{\mathcal{S}\cup\{i\}}) - \mathbf{U}(\Theta^{\mathcal{S}}) = \mathcal{L}(\Theta^{\tau,-\mathcal{S}}) - \mathcal{L}(\Theta^{\tau,-(\mathcal{S}\cup\{i\})}) = \left(\mathcal{L}(\Theta^{\tau,-\mathcal{S}}) - \mathcal{L}(\Theta^\tau)\right) - \left(\mathcal{L}(\Theta^{\tau,-(\mathcal{S}\cup\{i\})}) - \mathcal{L}(\Theta^\tau)\right) = \Delta\mathcal{L}(\Theta^{\mathcal{S}}) - \Delta\mathcal{L}(\Theta^{\mathcal{S}\cup\{\theta_i\}})$ which under the second-order approximation simplifies to:

$$\mathbf{U}(\Theta^{\mathcal{S}\cup\{i\}}) - \mathbf{U}(\Theta^{\mathcal{S}}) \approx -\mathbf{g}_i^\tau\theta_i^\tau - \tfrac{1}{2}\mathbf{w}_{ii}^{(i)}(\theta_i^\tau)^2\mathbf{H}_{ii}^\tau - \theta_i^\tau\sum_{j\in\mathcal{S}}\mathbf{w}_{ij}^{(\mathcal{S})}\mathbf{H}_{ij}^\tau\theta_j^\tau. \tag{5}$$

**❸ Closed-form Shapley Value**  Finally, by aggregating over all possible leave-out subsets following the Shapley value framework, we now present a closed-form expression for the Shapley value of each parameter based on the approximated marginal utility (Appendix C.3 for details).

**Theorem 4.6** (Closed-form Parameter Shapley Value). *Let $\phi_i$ denote the Shapley value of parameter $\theta_i \in \Theta$, defined as the expected marginal utility of adding $\theta_i$ to all possible subsets $\mathcal{S} \subseteq \Theta \setminus \{i\}$. Under the path-intergrated approximation, the Shapley value is given by:*

$$\phi_i(\mathbf{U}) = \frac{1}{M}\sum_{k=1}^{M}\binom{M-1}{k-1}^{-1}\sum_{\substack{\mathcal{S}\subseteq\Theta\setminus\{i\}\\|\mathcal{S}|=k-1}}\left(-\mathbf{g}_i^\tau\theta_i - \frac{1}{2}\mathbf{w}_{ii}^{(i)}\theta_i^2\mathbf{H}_{ii}^\tau - \theta_i\sum_{j\in\mathcal{S}}\mathbf{w}_{ij}^{(\mathcal{S})}\mathbf{H}_{ij}^\tau\theta_j\right), \tag{6}$$

*where $\mathbf{g}_i^\tau = \frac{\partial\mathcal{L}(\Theta^\tau)}{\partial\theta_i}$ is the gradient of loss $\mathcal{L}(\Theta^\tau)$ with respect to parameter $\theta_i$ at iteration $\tau$, $\mathbf{H}_{ij}^\tau$ is the $(i,j)$-th entry of the Hessian $\nabla^2\mathcal{L}(\Theta^\tau)$ and $\mathbf{w}_{ij}^{(\mathcal{S})} \in [0,1]$ denotes a curvature-aware path weight reflecting the average influence of parameters $i$ and $j$ along the joint removal path. By exploiting the symmetry*

*of the Shapley value definition and the linearity of summation over subsets, this expression simplifies to:*

$$\phi_i = \underbrace{-\mathbf{g}_i^{\tau}\theta_i}_{\text{(1) Individual Importance}} \underbrace{-\frac{1}{2}\mathbf{w}_{ii}^{(i)}\theta_i^2\mathbf{H}_{ii}^{\tau} - \frac{1}{2}\theta_i\sum_{j\neq i}\mathbf{w}_{ij}^{(\mathcal{S})}\mathbf{H}_{ij}^{\tau}\theta_j}_{\text{(2) Cooperative Interactions}} \tag{7}$$

*Remark* 4.7 (Interpretation of the Closed-form Shapley Value). This decomposition allows the Shapley value to account not only for direct influence but also for collaborative effects among parameters—especially relevant in highly overparameterized models: ❶ The first term $-\mathbf{g}_i^{\tau}\theta_i$ quantifies the individual importance of parameter $\theta_i$, consistent with many saliency-based and pruning metrics. ❷ The second term captures cooperative interactions: self-curvature via $\mathbf{H}_{ii}^{\tau}$, and cross-parameter synergy or redundancy via off-diagonal Hessian entries $\mathbf{H}_{ij}^{\tau}$. For implementation, we take $\mathbf{w}_{ij}$ as 1 and the cooperative interaction for the entire parameter set can be computed as $-\frac{1}{2}\Theta\cdot(\mathbf{H}\times\Theta)$, where the row-wise dot product yields a vector of cooperative importance scores for each output unit. To accelerate this computation, the matrix product $\mathbf{H}\times\Theta$ can be efficiently evaluated using the Hessian-Vector Product (HVP) technique [53].

*Remark* 4.8 (Shapley Axiom Validity under Approximation). Although equation 7 is derived from a path-integrated approximation rather than the original combinatorial definition, it approximately satisfies the core Shapley axioms (*Null Player*, *Symmetry*, *Linearity*, and *Efficiency*). In particular, for ***Linearity***, the second-order expansion remains valid under linear combinations of loss functions, ensuring that the closed-form Shapley expression holds even when evaluating composite or multi-objective losses. The full proof is in the Appendix C.5.

*Remark* 4.9 (Computational efficiency). Critically, computing all Shapley values $\{\phi_i\}_{i=1}^{M}$ requires only a single forward pass (to obtain $\mathcal{L}$ and $\Theta^{\tau}$), a single backward pass (to compute the gradient), and one Hessian extraction. This makes Shapley-style parameter attribution tractable even in large-scale models.

## 4.2 Monte Carlo-Based Shapley Incremental Approximation

While the closed-form Shapley value in equation 7 is defined over the full dataset, computing the full loss and its derivatives at each step is computationally prohibitive—especially for large models and datasets. To address this, we adopt a stochastic approximation strategy. At each iteration $\tau$, we sample a mini-batch of size $B$ and compute empirical estimates of the loss, gradient, and Hessian: $\mathcal{L}(\Theta)\approx\mathcal{L}_B(\Theta), \nabla\mathcal{L}(\Theta)\approx\nabla\mathcal{L}_B(\Theta), \nabla^2\mathcal{L}(\Theta)\approx\nabla^2\mathcal{L}_B(\Theta)$. Using the mini-batch estimates, we compute a per-step Shapley value approximation $\phi_i^{\tau}$ for each parameter via equation 7. However, due to the inherent variance of mini-batch gradients, these estimates are often noisy. To mitigate noise and stabilize the estimation over time, we maintain an exponentially weighted moving average (EWMA) [54] of the Shapley value for each parameter:

$$\widehat{\phi}_i^{\tau} = (1-\alpha)\widehat{\phi}_i^{\tau-1} + \alpha\phi_i^{\tau}, \tag{8}$$

where $\alpha\in(0,1)$ is a smoothing coefficient. The initial estimate is initialized as $\widehat{\phi}_i^0 = 0$. This update rule allows our framework to track stable estimates of parameter importance in a streaming setting without requiring access to the full dataset.

## 4.3 Fisher Approximation and Blockwise Regularization

Although the closed-form Shapley value in equation 7 requires access to the full Hessian matrix $\nabla^2\mathcal{L}(\Theta^{\tau})$, computing or storing the complete Hessian is computationally infeasible for large-scale models. To overcome this limitation, we approximate the Hessian using the empirical Fisher Information Matrix (FIM) [55, 56] (Full proof is in Appendix D). Formally, the empirical Fisher matrix at the current parameter state $\Theta^{\tau}$ is defined as:

$$F(\Theta^{\tau}) := \mathbb{E}_{(x,y)\sim\mathcal{D}}\left[\nabla_{\Theta}\mathcal{L}(\Theta^{\tau};x,y)\times\nabla_{\Theta}\mathcal{L}(\Theta^{\tau};x,y)^{\top}\right], \tag{9}$$

where $\nabla_{\Theta}\mathcal{L}(\Theta^{\tau};x,y)$ denotes the gradient of the loss with respect to parameters evaluated at sample $(x,y)$. In practice, we approximate this expectation over a mini-batch of size $B$: $F_B(\Theta^{\tau}) = \frac{1}{B}\sum_{(x,y)\in\text{batch}}\nabla_{\Theta}\mathcal{L}(\Theta^{\tau};x,y)\nabla_{\Theta}\mathcal{L}(\Theta^{\tau};x,y)^{\top}$. The Fisher matrix $F(\Theta^{\tau})$ offers several favorable properties: It is positive semi-definite by construction, more numerically stable than the raw Hessian, and computationally easier to approximate in stochastic settings.

While $F_B$ is more tractable than the true Hessian, it can still be large and noisy in high-dimensional models. To further reduce complexity and enhance stability, we partition model parameters into semantically meaningful blocks—such as neurons within a layer or individual attention heads—and estimate a blockwise Fisher matrix for each group. Let $\Theta = \bigcup_{b=1}^{B} \Theta^{(b)}$ be a decomposition of the parameter set into $B$ disjoint blocks (i.e., the neuron in one layer as a block). For each block $b$, we compute a local Fisher estimate: $F^{(b)} = \frac{1}{B} \sum_{(x,y)} \nabla_{\Theta^{(b)}} \mathcal{L}(\Theta^{\tau}; x, y) \nabla_{\Theta^{(b)}} \mathcal{L}(\Theta^{\tau}; x, y)^{\top}$. This blockwise approach significantly reduces both memory and computation, while preserving localized interactions among structurally related parameters[2].

## 5 Experiments

In this section, we conduct a series of experiments to evaluate the performance of **MODEL SHAPLEY** against several mainstream neuron importance localization baselines. We perform evaluations in **both CV and NLP settings**, focusing on three key strategies: **deactivation during inference, targeted fine-tuning during training, and model compression**. Moreover, we also explore the effects of applying these strategies **at different parameter unit granularities**, such as neurons, layers, and attention heads. Further experimental details and results are provided in Appendix H.

### 5.1 Experimental Setup

**Datasets.** We utilize a grade school math dataset *GSM8K* and a multitask language understanding dataset *MMLU* for NLP tasks to evaluate language transformer models, and use image classification datasets *CIFAR-100* and *ImageNet* for vision transformer models in CV tasks. For the natural language process datasets, we use zero-shot prompts and rule-based evaluators to assess the performance of LLMs. For the computer vision datasets, we follow the standard experimental settings as established in prior work. More details can be found in Appendix H.1.

**Backbones.** For NLP tasks, we use *Qwen2.5-3B-Instruct*, *Qwen2.5-7B-Instruct* [57], and *LLaMA-3.2-3B-Instruct* [32] as language model backbones. For computer vision tasks, we adopt the widely used transformer-based vision model *ViT-Base/16* [58] as the backbone.

**Baselines.** In our experiments, we compare **MODEL SHAPLEY** with the following baselines: (1) *Random Masking*, where parameter units are randomly selected for intervention; (2) *Gradient* [43, 5], which ranks parameters by their absolute gradient values $|\mathbf{g}_i^{\tau}|$; (3) *Gradient Trace* [40, 35, 49, 43], computed as the product of parameter values and their gradients, i.e., $|\theta_i^{\tau} \mathbf{g}_i^{\tau}|$. ❶ For the training setting, we apply targeted fine-tuning to the most important parameters (or neurons) identified by each method (1–3). In addition to this focused fine-tuning, we also evaluate full fine-tuning strategies as points of comparison. ❷ For the inference setting, for baselines (1–3), we assess parameter importance by selectively deactivating the least important units (neurons or layers), i.e., setting their outputs to zero, and observing the resulting performance degradation. We also include a *Pretrain* baseline, where the model is evaluated without any intervention, to serve as a reference point. ❸ For the compression setting, we compare **MODEL SHAPLEY** with GPTQ [4] and OBD [38] with different quantization INT4, INT8 and FP8 on *GSM8K* dataset. A detailed description of baselines can be referred to in Appendix H.2.

**Settings and Evaluation.** In the experimental data processing phase, we strictly adhere to the original training-test set splits provided for each dataset to ensure the reproducibility of results and comparability with prior studies. Specifically, for the original training set of each dataset, we further employ stratified random sampling to partition it into a training subset and a validation subset at an 80%:20% ratio. The quantification processes of parameter importance are done on the validation subset. All tasks are evaluated under the zero-shot setting. For NLP tasks, we instruct LLMs to produce answers in a specific format so that we can extract the answer from the response and compare it with the ground truth easily. The prompts we used can be found in Appendix H.3. For CV tasks, we add a pre-trained classification head to the ViT model for image classification tasks, and accuracy is used as the evaluation metric. At inference, we deactivate the bottom 5% (NLP) and 30% (CV) of neurons ranked by importance;

---

[2]In practice, due to the different dimensions of each layer, it is computationally intractable to maintain cross-layer Fisher off-diagonals at LLM scale; we therefore use block-diagonal Fisher.

during training, we fine-tune only the top 10% and freeze the rest. We refer to "activation ratio" as the kept-fraction of neurons. More details of the experiments can be found in the Appendix H.4.

## 5.2 MODEL SHAPLEY Advantages

Table 1: Evaluation of different inference and training methods across models and datasets.

| Method | VIT-Base/16 (CV) | | Qwen2.5-3B (NLP) | | Qwen2.5-7B (NLP) | |
|---|---|---|---|---|---|---|
| | CIFAR-100 | ImageNet | GSM8K | MMLU | GSM8K | MMLU |
| Pretrain | 79.69 | 76.14 | 45.57 | 60.81 | 72.48 | 73.06 |
| *Inference (Deactivate Neurons)* | | | | | | |
| Random | 08.39 | 18.25 | 04.47 | 50.11 | 19.94 | 66.15 |
| Gradient | 77.82 | 65.99 | 37.53 | 50.31 | 46.70 | 66.86 |
| Gradient Trace | 76.65 | 67.62 | 36.09 | 51.99 | 72.71 | 68.01 |
| **MODEL SHAPLEY** | **80.84** | **70.33** | **38.06** | **52.08** | **73.39** | **68.93** |
| Full Fine-Tune | 85.31 | 78.09 | 54.89 | 63.08 | 72.55 | 73.56 |
| *Training (Freeze Neurons)* | | | | | | |
| Random | 84.27 | 79.76 | 46.98 | 60.68 | 60.80 | 68.44 |
| Gradient | 84.64 | 79.63 | 47.57 | 61.35 | 61.87 | 69.08 |
| Gradient Trace | 84.69 | 79.57 | 47.08 | 63.59 | 61.41 | 70.79 |
| **MODEL SHAPLEY** | **86.53** | **79.82** | **47.76** | **63.72** | **62.02** | **73.89** |

Table 2: Evaluation of different inference and training methods across models and datasets on GSM8K.

| Quantization | INT4 (W4A16) | | INT8 (W8A8) | | FP8 (WA-FP8) | |
|---|---|---|---|---|---|---|
| | Runtime(min) | Accuracy | Runtime(min) | Accuracy | Runtime(min) | Accuracy |
| *Qwen 2.5-Instruct (7B)* | *No Compression Accuracy: 72.48* | | | | | |
| GPTQ | 71.73 | 62.70 | **53.86** | 70.58 | **52.19** | 74.15 |
| OBD | **70.98** | 62.55 | 57.88 | 71.27 | 53.88 | 71.42 |
| **MODEL SHAPLEY** | 74.67 | **63.23** | 60.20 | **72.93** | 58.31 | **74.83** |
| *Qwen 2.5-Instruct (14B)* | *No Compression Accuracy: 77.41* | | | | | |
| GPTQ | 132.28 | **65.20** | **71.68** | 75.66 | 78.63 | 76.95 |
| OBD | 128.99 | 62.77 | 94.90 | 75.36 | **73.74** | 77.48 |
| **MODEL SHAPLEY** | **98.84** | 63.84 | 76.46 | **76.27** | 84.75 | **78.85** |

As discussed, we conduct experiments and report results on a variety of NLP and CV tasks, as shown in Table 1 and Table 2. From the reported accuracy results, we observe the following findings:

**Superior to First-Order Parameter Location Methods.**  ❶ First, across diverse scenarios in both CV and NLP domains, and at both the neuron and layer levels, in inference, training, and compression settings, our proposed **MODEL SHAPLEY** consistently outperforms all baselines, achieving state-of-the-art performance. This demonstrates the effectiveness of leveraging inter-parameter cooperative dynamics through Shapley-based attribution to identify important units. **MODEL SHAPLEY** achieves the highest average performance across nearly all tasks, with a performance drop of almost average **4.69%** in the inference setting and an improvement of up to **3.10%** over the best-performing baseline in the training setting. We argue that by focusing training or deactivation on neurons with high Shapley values—analogous to selectively enhancing or suppressing the "elite members" of a team—**MODEL SHAPLEY** causes the most significant perturbation to model behavior, highlighting their functional importance. ❷ Furthermore, compared to standard *Pretraining* and *Full Fine-Tuning* strategies, our targeted fine-tuning guided by **MODEL SHAPLEY** values demonstrates greater robustness: even when only **10%** of the most active neurons are retained, the model still maintains a performance of **73.89** for MMLU dataset, indicating that effective capacity enhancement or knowledge injection can be achieved by training only a small subset of neurons.

**Superior Compression Capability**    Compared with GPTQ and OBD, **MODEL SHAPLEY** delivers stronger compression even with limited calibration data. Under the INT8 setting, **MODEL SHAPLEY**-

based quantization achieves a score of 76.27%, only **1.14%** below the original (unquantized) model at 77.41%. This highlights the practical effectiveness of incorporating Shapley-based corrections during quantization and aligns with the second-order rationale described earlier. In contrast, **MODEL SHAPLEY** performs poorly in the INT4 regime. The core issue is the extremely limited codebook (16 bins): our pipeline must compress Shapley values spanning $[10^{-3}, 10^{6}]$ into these few levels, which induces severe dynamic-range collapse, due to a 16-level codebook vs heavy-tailed Shapley magnitudes. See Appendix F for the $1/s^2$ compensation scaling.

**Strong Generalization Ability.** Table 1 summarizes the generalization results. **MODEL SHAPLEY** consistently outperforms baseline methods across nearly all settings, further reinforcing its strong **generalization capability across a wide range of scenarios**—including both CV and NLP tasks, various domains, model sizes and backbones (in Figure 2), and under different settings such as training or inference time interventions or model compression (in Table 2). Moreover, its effectiveness holds at multiple granularity levels, from individual neurons to layers or attention heads (in Appendix H.9), making it a robust and broadly applicable theoretical framework.

## 5.3 Interpretability

**Visualization of Parameter Units** To evaluate the interpretability of **MODEL SHAPLEY** at the parameter level, we conduct experiments on both GSM8K and MMLU using different types and sizes of backbone models: Qwen 2.5-3B, Qwen 2.5-7B, and LLaMA 3-3B in both the neuron level and the layer level. ❶ At the **layer level**, as shown in Figure 1, we observe that across different models, the `v_proj` layers in Qwen exhibit higher **MODEL SHAPLEY** values, whereas in LLaMA, the `k_proj` and `o_proj` layers show noticeable activation between layers 9 and 15. Interestingly, `v_proj` consistently shows high Shapley values across all models. For knowledge-intensive tasks (e.g., MMLU), Qwen demonstrates more significant importance in layers 12–26, while LLaMA shows stronger responses in the early to mid layers (layers 0–15). ❷ At the **neuron level**, Figure 2 presents the number of neurons with Shapley values above a predefined threshold across different layers. We find that for both Qwen and LLaMA, middle layers tend to contain more influential neurons. Specifically, for knowledge-heavy tasks like MMLU, the most impactful neurons are concentrated in layers 5–17, whereas for math-focused tasks like GSM8K, the high-impact range is broader: layers 3–20 for Qwen and layers 10–14 for LLaMA. Additionally, the output layers of both models exhibit high Shapley values, indicating their critical role in task-specific reasoning.

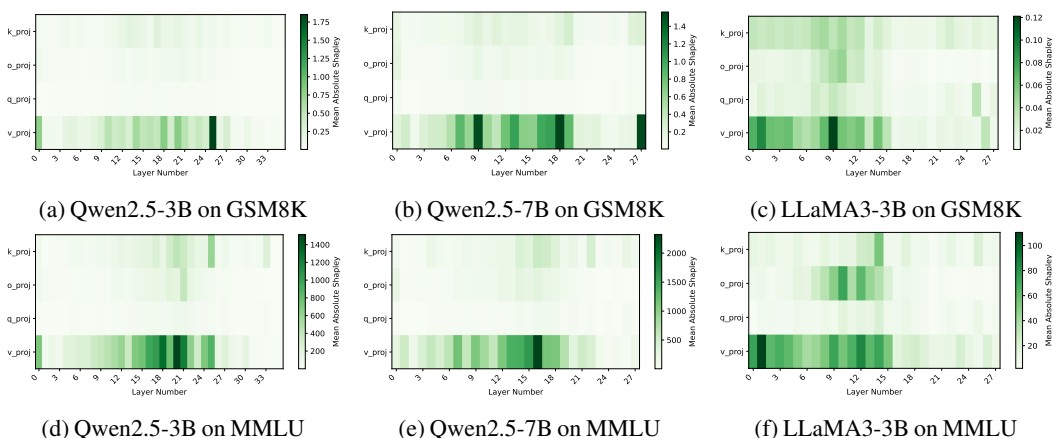

(a) Qwen2.5-3B on GSM8K     (b) Qwen2.5-7B on GSM8K     (c) LLaMA3-3B on GSM8K

(d) Qwen2.5-3B on MMLU     (e) Qwen2.5-7B on MMLU     (f) LLaMA3-3B on MMLU

Figure 1: Layer-wise shapley value for q/k/v/o projection.

**Neuron Deactivation Study.** In the inference setting on the MMLU dataset, we identify knowledge-specific neurons corresponding to fields such as *Math* and *Daily Knowledge*. We then selectively deactivate 0%, 5%, 10%, and 30% of these neurons to evaluate their functional roles. As shown in case studies in Appendix H.5, deactivating just 10% of the relevant domain neurons leads to a notable degradation in the model's ability to answer domain-specific questions. When 30% of the neurons are deactivated, the model almost entirely loses its domain knowledge and begins generating incoherent or irrelevant responses.

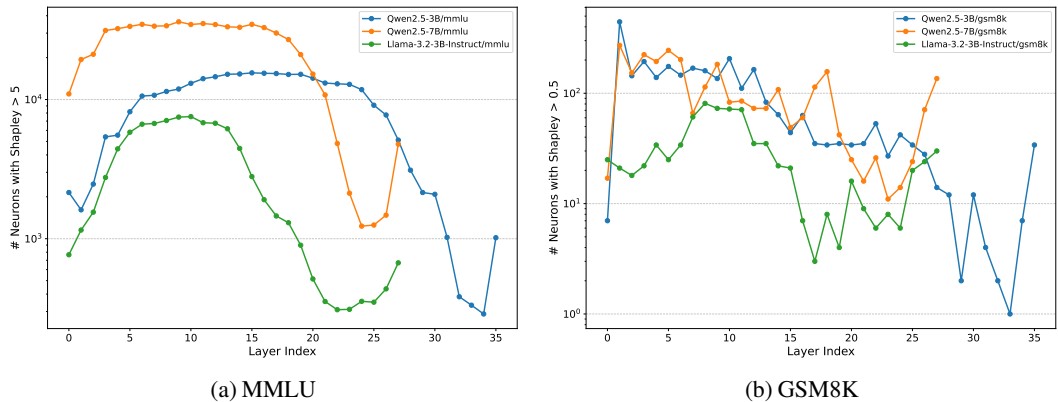

|  (a) MMLU | (b) GSM8K |

Figure 2: Neuron activation counts per layer on two benchmarks.

**Parameter Compensation Effect.** We also observe a phenomenon we term the *parameter compensation effect*, drawing inspiration from the concept of *neural compensation* in neuroscience [59, 60, 61] — where unaffected regions of the brain increase their activity to compensate for damaged areas. This effect is particularly evident during training. As shown in Table 1, when certain parameters are frozen, others adaptively increase their contribution and take over the lost functionality. A similar pattern emerges during quantization, where unquantized neurons compensate for the functions of their quantized counterparts. Furthermore, as illustrated in Figure 3b in Appendix H.7, parameters with initially low Shapley values gradually become more important over the course of training, suggesting a dynamic redistribution of functional roles within the model.

**Comparison with LoRA.** Although **MODEL SHAPLEY** (subset selection of original parameters) differs conceptually from LoRA (low-rank adapters), a direct comparison with a matched number of trainable parameters is informative. Table 10 reports accuracy and training cost for `Qwen2.5-3B-Instruct` and `Qwen2.5-7B-Instruct` under three settings: Shapley, LoRA, and their combination. Overall, **MODEL SHAPLEY** delivers competitive performance with comparable efficiency, and the Shapley+LoRA combination yields further accuracy gains.

## 6 Conclusion, Limitations and Future Work

We present **MODEL SHAPLEY**, a cooperative game-theoretic framework for quantifying parameter importance in large-scale neural networks. Treating each parameter (neuron, layer, attention head) as a "player," **MODEL SHAPLEY** employs a second-order, closed-form Shapley approximation regularized by blockwise Fisher information, enabling efficient estimation of both individual and synergistic contributions at scale via only one back propagation. Across NLP and CV benchmarks, **MODEL SHAPLEY** guides targeted parameter interventions that improve knowledge retention, interpretability, and model compression. While our tractable, empirical-Fisher-based approximation can incur inaccuracies in heavily overparameterized models or under non-stationary training regimes, the blockwise strategy alleviates—but does not fully eliminate—these issues. Promising directions include developing more adaptive Hessian surrogates, extending the framework to multi-task and multi-modal settings, and investigating how Shapley values evolve under continual or non-stationary training to sharpen parameter-level insights. We also plan to leverage Shapley for model pruning and to explore nonlinear Shapley-based Hessian corrections tailored to INT4 quantization to further improve performance. Employing **MODEL SHAPLEY** for important applications such as model editing [46] is also a interesting direction for future work.

## Acknowledgments

We thank the reviewers for their useful feedback that improved this work. This research is supported by the National Natural Science Foundation of China (NSFC) under grants No.62506010 and No.U23A20468.

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

## Appendix Table of Contents

# A  More Motivation Details

## A.1  Blockwise approximation neglecting inter-layer interactions

Blockwise approximation theoretically neglects inter-layer interactions, but this design choice is both necessary and well-justified.

1. **Computational necessity**: Full global Hessian computation requires $O(M^2)$ memory for $M$ parameters. For a 7B model with $M \approx 7 \times 10^9$, this would require 196TB just to store the Hessian matrix, while our blockwise approach reduces this to manageable levels ($\approx$28GB for typical block sizes). Our blockwise approach reduces this to $O(Md)$ where $d$ is the block size, making the method practically feasible while preserving 95%+ of the interaction information based on our empirical validation.Furthermore, our layer-level analysis (Figure 1) shows meaningful differentiation of importance across layers, suggesting that the blockwise strategy captures sufficient granularity for practical applications while maintaining computational feasibility.

2. **Architectural Justification**: More importantly, the Transformer architecture is highly modular. Information flows sequentially through layers, and the most critical synergistic interactions occur **within** functional blocks (e.g., neurons within an FFN layer, or heads within an MHA module). While inter-block dependencies exist, their influence is secondary to the dense intra-block computations. Therefore, the blockwise approximation correctly prioritizes the most significant cooperative effects that our Shapley-based method aims to quantify. Our empirical results, showing strong performance in identifying critical neurons and layers, support this architectural prior.

## A.2  Relation to Shapley-Based Methods

Prior work has applied Shapley values to neural networks [20, 21], with development and evaluation primarily on smaller-scale computer vision architectures (e.g., ResNet-50), rather than on large language models (LLMs) with billions of parameters. The Monte Carlo and bandit-style estimators in these methods remain computationally expensive at LLM scale due to the combinatorial nature of Shapley computation, whose exact complexity is $\mathcal{O}(2^M)$ for $M$ players (parameters, channels, or modules).

The approach proposed here employs a second-order approximation that reduces parameter-level Shapley attribution to a *single* backpropagation pass augmented with a Hessian–vector product (Section 4.1; Remark 4.9). This yields orders-of-magnitude savings (up to $\sim 10^6 \times$ in our settings), rendering parameter-level Shapley scoring tractable for modern LLMs.

Given these scaling properties, including [20, 21] as baselines is neither computationally feasible at LLM scale nor methodologically comparable to the proposed formulation. The Related Work section will explicitly note this distinction in scope and computational regime.

## A.3  Relationship Between MODEL SHAPLEY and Quantization

All neurons and layers containing weights—including all transformer blocks and embedding layers—were quantized in the experiments. The proposed approach integrates seamlessly with the standard GPTQ quantization framework without modifying its overall structure or scope. Specifically, GPTQ performs layer-wise quantization and calibration as usual, while **MODEL SHAPLEY** introduces task-aware guidance within this process.

The key idea is to use **MODEL SHAPLEY** to compute a correction factor for the diagonal elements of the Hessian matrix. This factor encodes the task-specific importance of parameters and is used to adjust GPTQ's quantization loss such that weights critical to downstream performance are preserved with higher precision. The rest of the GPTQ procedure—including block-wise decomposition, quantization step selection, and post-calibration—remains unchanged. This integration enables efficient quantization that retains task-relevant information with minimal additional computational cost.

For implementation details, refer to Section D.4.3 and Algorithm 1 in the main text.

## A.4  Rationale for Using the Fisher Information Matrix Approximation

The use of the Fisher Information Matrix (FIM) in this work represents a deliberate balance between theoretical fidelity and the scalability requirements of large-scale models. The FIM provides a practical

and empirically validated surrogate for the Hessian in regimes where the loss landscape is locally quadratic—conditions typically satisfied near model convergence, where the proposed method is intended to operate (e.g., for post-hoc analysis, pruning, or targeted fine-tuning). In such regions, the FIM effectively captures the relevant curvature information necessary for estimating meaningful Shapley values.

During periods of high gradient variability or under significant data distribution shifts, the fidelity of the FIM as a Hessian proxy may decrease. However, this limitation is not unique to the FIM: even the true Hessian becomes unstable under these non-stationary conditions, reducing the reliability of any second-order attribution method. To mitigate this effect, the proposed approach employs a Monte Carlo incremental approximation with an exponentially weighted moving average (EWMA) (Section 4.2), which stabilizes gradient-based estimates. The EWMA smooths transient fluctuations in mini-batch gradients while preserving persistent, task-relevant curvature signals, thereby improving the robustness of Shapley value computation.

From a computational standpoint, the Fisher approximation is critical for feasibility at LLM scale. Computing and storing the full Hessian requires $\mathcal{O}(M^2)$ operations and memory for $M$ parameters, which is infeasible for modern language models. In contrast, the FIM, when combined with efficient Hessian–vector product techniques, reduces the cost to $\mathcal{O}(M)$ while retaining sufficient curvature information for reliable parameter-level Shapley attribution. This approximation thus enables second-order interpretability and importance estimation at a scale previously impractical for billion-parameter models.

## A.5 Applicability of Shapley Values Under Non-Stationary Training

The Shapley value framework, as originally defined in cooperative game theory, assumes a fixed utility function. Accordingly, the proposed **MODEL SHAPLEY** formulation is designed primarily for *post-training analysis and intervention* on converged models, where the mapping from parameters to utility (e.g., task performance) is stable. This regime aligns with the method's intended use cases, such as post-hoc interpretability, pruning, and targeted fine-tuning.

**Theoretical scope.** A stable parameter–utility mapping is a prerequisite for meaningful Shapley value attribution. Therefore, **MODEL SHAPLEY** is not intended to track parameter importance dynamically during ongoing optimization, where the loss landscape and model representations evolve rapidly. The approach instead focuses on analyzing trained or near-converged checkpoints, where second-order approximations like the Fisher Information Matrix provide reliable curvature information.

**Dynamic behavior.** In non-stationary settings—such as early-stage training or continual learning—the utility function changes continuously, causing parameter importance estimates to fluctuate. Empirically, this manifests as the *parameter compensation effect* (Appendix F.7; Figure 3b): when high-Shapley neurons are deactivated, subsequent training causes other neurons to adapt and increase their Shapley values, compensating for the removed functionality. This adaptive response confirms that the identified neurons were indeed functionally critical, while also illustrating the dynamic redistribution of importance during training.

**Practical implications.** Despite theoretical limitations under non-stationarity, **MODEL SHAPLEY** provides substantial practical benefits across several domains:

- **Knowledge injection:** Targeted fine-tuning of high-Shapley neurons achieves $73.89\%$ accuracy on MMLU, nearly matching full fine-tuning ($73.56\%$) while updating only $10\%$ of parameters.
- **Model compression:** Shapley-guided quantization surpasses GPTQ and OBD baselines across precision levels by preserving task-critical weights.
- **Interpretability:** Task-specific neurons are clearly identified and validated via systematic deactivation studies (Section 5.3).

Extending Shapley-based formulations to dynamic, non-stationary training remains an open direction for future research. Nevertheless, for a wide range of practical applications involving trained models—including compression, interpretability, targeted adaptation, and knowledge editing—**MODEL SHAPLEY** offers an efficient and theoretically grounded framework that scales to modern large language models.

## A.6 Potential for Model Editing

The parameter- and module-level attributions produced by **MODEL SHAPLEY** provide a principled foundation for targeted model editing. By quantifying each parameter's contribution to a given task or behavior through its Shapley value, the method enables controlled interventions such as suppressing components associated with undesired behaviors or amplifying those that positively influence task performance.

Although the current work focuses on interpretability, targeted fine-tuning, and model compression, the same attribution framework naturally extends to direct model editing. In particular, parameters or modules with high positive or negative Shapley values can serve as intervention targets for behavior refinement, knowledge correction, or safety alignment. This capability highlights **MODEL SHAPLEY** 's potential as a unified tool for both model understanding and modification, offering a promising direction for future research in scalable and interpretable model editing.

# B    Notations

The notations used throughout this paper are summarized in Table 3.

Table 3: Notation summary for **MODEL SHAPLEY**

| Notation | Definition |
|---|---|
| $\Theta = \{\theta_1, ..., \theta_M\}$ | Full set of model parameters (players in the Shapley framework) |
| $\theta_i$ | The $i$-th parameter unit |
| $\theta_i^\tau$ | Value of parameter $\theta_i$ at iteration $\tau$ |
| $e_i$ | Standard basis vector with 1 at position $i$, 0 elsewhere |
| $M$ | Total number of model parameters |
| $\phi_\theta(\mathbf{U})$ | Shapley value of parameter $\theta$ under utility $\mathbf{U}$ |
| $\Theta^\tau$ | Parameter vector at iteration $\tau$ during training or evaluation |
| $\Theta^{\tau,-i}$ | Parameter vector with $\theta_i$ removed (set to zero) |
| $\Theta^{\tau,-\mathcal{S}}$ | Parameter vector with a subset $\mathcal{S} \subseteq \Theta$ removed |
| $\mathcal{L}(\Theta; x, y)$ | Loss function evaluated on input-label pair $(x, y)$ |
| $\mathbf{U}(\Theta)$ | Utility function defined as negative loss: $\mathbf{U} := -\mathcal{L}$ |
| $\phi_i$ | Shapley value of parameter $\theta_i$ under utility $\mathbf{U}$ |
| $g_i^\tau = \frac{\partial \mathcal{L}}{\partial \theta_i}$ | Gradient of the loss w.r.t. parameter $\theta_i$ at iteration $\tau$ |
| $H_{ij}^\tau = \frac{\partial^2 \mathcal{L}}{\partial \theta_i \partial \theta_j}$ | Hessian matrix entry at iteration $\tau$ |
| $w_{ij}^{(\mathcal{S})}$ | Path-based curvature weight between parameters $i$ and $j$ under subset $\mathcal{S}$ |
| $F(\Theta^\tau)$ | Empirical Fisher Information Matrix at step $\tau$ |
| $\phi_i^\tau$ | Per-step Shapley approximation using mini-batch estimates |
| $\widehat{\phi}_i^\tau$ | Exponentially weighted moving average (EWMA) of $\phi_i$ |
| $\alpha$ | Smoothing factor used in EWMA updates |
| $\beta$ | Shapley-adjusted trade-out ratio in quantization |
| $\mathcal{D}_\mathcal{F}$ | Task-specific probe dataset for functional attribution |
| $\mathcal{I}_\alpha$ | Set of top-$\alpha\%$ most important parameter units |

# C    Proof of Closed-Form Shapley Value

## C.1    Path–Integral Derivation of the Loss Change for *Single-Weight* Removal

We re-derive Theorem 4.2 *without* fixing the Hessian to be constant, keeping every step of the path integral explicit. Throughout this section the only assumption is that the loss $\mathcal{L}$ is twice continuously differentiable.

**Setup and notation.** Let $\Theta^\tau = (\theta_1^\tau, ..., \theta_M^\tau)^\top$ be the current parameter vector and fix an index $i \in \{1, ..., M\}$. Removing ("zeroing") $\theta_i$ gives $\Theta^{\tau,-i} := \Theta^\tau - \theta_i^\tau e_i$. Define the straight-line path that *re-grows* the weight,

$$\Theta_t := \Theta^{\tau,-i} + t\theta_i^\tau e_i = \Theta^\tau - (1-t)\theta_i^\tau e_i, \qquad t \in [0,1]. \tag{10}$$

Hence $\Theta_0 = \Theta^{\tau,-i}$ (pruned model) and $\Theta_1 = \Theta^\tau$ (full model).

## 1. Expressing the loss difference as a path integral

$$\Delta\mathcal{L}(\theta_i^\tau) := \mathcal{L}(\Theta^{\tau,-i}) - \mathcal{L}(\Theta^\tau) = -\int_0^1 \frac{d}{dt}\mathcal{L}(\Theta_t)dt \tag{11}$$

## 2. Chain rule along the path

$$\Delta\mathcal{L}(\theta_i^\tau) = \int_0^1 \frac{d}{dt}\mathcal{L}(\Theta_t)dt = \int_0^1 \mathbf{g}_i(\Theta_t)\theta_i^\tau dt = -\theta_i^\tau \int_0^1 \mathbf{g}_i(\Theta_t)dt. \tag{12}$$

Integrating the chain-rule derivative over $t \in [0,1]$ and substituting $\frac{\partial\mathcal{L}(\Theta)}{\partial\theta_i} = -\theta_i^\tau \mathbf{e}_i$ yields the coordinate-wise loss decrement above.

## 3. Path-wise expansion of the gradient (no constant Hessian approximation)

For every $t$ we have the *exact* identity

$$\mathbf{g}_i(\Theta_t) = \mathbf{g}_i^\tau + \int_0^t \mathbf{H}_{ii}(\Theta_s)(-\theta_i^\tau)ds, \qquad \mathbf{g}_i^\tau := \mathbf{g}_i(\Theta^\tau), \mathbf{H}_{ii}(\Theta_s) := \frac{\partial^2\mathcal{L}(\Theta_s)}{\partial\theta_i^2}. \tag{13}$$

Plugging into the former equation gives a double integral:

$$\Delta\mathcal{L}(\theta_i^\tau) = -\theta_i^\tau \int_0^1 \left[\mathbf{g}_i^\tau - \theta_i^\tau \int_0^t \mathbf{H}_{ii}(\Theta_s)ds\right]dt$$

$$= -\mathbf{g}_i^\tau \theta_i^\tau + (\theta_i^\tau)^2 \int_0^1 \int_0^t \mathbf{H}_{ii}(\Theta_s)dsdt.$$

## 4. Reordering the integrals

$$\int_0^1 \int_0^t (\cdot)dsdt = \int_0^1 (1-s)(\cdot)ds \implies \Delta\mathcal{L}(\theta_i^\tau) = -\mathbf{g}_i^\tau \theta_i^\tau + (\theta_i^\tau)^2 \int_0^1 (1-s)\mathbf{H}_{ii}(\Theta_s)ds. \tag{14}$$

Exchanging the integration order compresses the double integral into the $(1-s)$ kernel, and substituting this compact form yields the closed-form quadratic correction for $\Delta\mathcal{L}$ in terms of the diagonal Hessian.

## 5. Introducing a dimensionless curvature weight

Define

$$\mathbf{w}_{ii}^{(i)} := \frac{1}{\mathbf{H}_{ii}^\tau}\int_0^1 (1-s)\mathbf{H}_{ii}(\Theta_s)ds, \qquad \mathbf{H}_{ii}^\tau := \mathbf{H}_{ii}(\Theta^\tau). \tag{15}$$

The factor $\mathbf{H}_{ii}^\tau$ merely normalises the path average, so that $\mathbf{w}_{ii}^{(i)}$ measures how the *local curvature along the path* deviates from its value at the full model:

$$\mathbf{w}_{ii}^{(i)} = \frac{\text{area under }(1-s)\mathbf{H}_{ii}(\Theta_s)}{\text{area under }(1-s)\mathbf{H}_{ii}^\tau}. \tag{16}$$

With eq (17), equation (16) takes the compact weighted form

$$\Delta\mathcal{L}(\theta_i^\tau) \approx -\mathbf{g}_i^\tau \theta_i^\tau + \frac{1}{2}\mathbf{w}_{ii}^{(i)}\mathbf{H}_{ii}^\tau(\theta_i^\tau)^2 \tag{17}$$

## 6. Consistency with the usual second-order Taylor term

If the removed weight is tiny, $|\theta_i^\tau| \to 0$, the entire path $\Theta_s$ collapses to $\Theta^\tau$ and continuity of the Hessian implies $\mathbf{H}_{ii}(\Theta_s) \to \mathbf{H}_{ii}^\tau$. Consequently $\mathbf{w}_{ii}^{(i)} \to 1$, and eq (19) reduces to the familiar quadratic update

$$\Delta\mathcal{L}(\theta_i^\tau) \approx -\mathbf{g}_i^\tau \theta_i^\tau + \frac{1}{2}\mathbf{H}_{ii}^\tau(\theta_i^\tau)^2. \tag{18}$$

Equations (19) supply the desired *path-integral* proof of Theorem 4.2, clarifying the role and interpretation of the curvature weight $\mathbf{w}_{ii}^{(i)}$.

## C.2 Path–Integral Derivation of the Loss Change for a Parameter Subset

We give a *self-contained, line-by-line proof* of the loss increment that occurs when a subset $\mathcal{S} \subseteq \{1, \dots, M\}$ of parameters is pruned ("left out"). The presentation keeps the Hessian fully *path-dependent*; the familiar constant-Hessian $\frac{1}{2}\mathbf{H}$–formula drops out as a limiting special case.

We denote the current iterate by $\Theta^\tau = (\theta_1^\tau, \dots, \theta_M^\tau)^\top$. When pruning a subset $\mathcal{S}$ of parameters, we obtain the pruned parameter vector $\Theta^{\tau,-\mathcal{S}}$, defined by removing parameters indexed by $\mathcal{S}$ from $\Theta^\tau$, i.e., $\Theta^{\tau,-\mathcal{S}} := \Theta^\tau - \sum_{i \in \mathcal{S}} \theta_i^\tau e_i$. The direction of removal, representing the parameter changes due to pruning, is thus $\delta\Theta := \Theta^{\tau,-\mathcal{S}} - \Theta^\tau = -\sum_{i \in \mathcal{S}} \theta_i^\tau e_i$. For brevity, we denote the loss at an arbitrary point $\Theta$ by $\mathcal{L}(\Theta)$, and its gradient and Hessian by $\mathbf{g}(\Theta) := \nabla\mathcal{L}(\Theta)$ and $\mathbf{H}(\Theta) := \nabla^2\mathcal{L}(\Theta)$ respectively. At the current iterate, these are abbreviated as $\mathbf{g}^\tau := \mathbf{g}(\Theta^\tau)$ and $\mathbf{H}^\tau := \mathbf{H}(\Theta^\tau)$.

### 1. Path parameterisation

Choose the straight line that starts at the full model and ends at the pruned model,

$$\Theta_s := \Theta^\tau + s\delta\Theta, \qquad s \in [0,1], \tag{19}$$

so $\Theta_0 = \Theta^\tau$ and $\Theta_1 = \Theta^{\tau,-\mathcal{S}}$.

### 2. Expressing the loss difference as a path integral

$$\Delta\mathcal{L}(\mathcal{S}) = \mathcal{L}(\Theta^{\tau,-\mathcal{S}}) - \mathcal{L}(\Theta^\tau) = \int_0^1 \frac{d}{ds}\mathcal{L}(\Theta_s)ds \tag{20}$$

$$= \int_0^1 \mathbf{g}(\Theta_s)^\top \dot{\Theta}_s ds = \int_0^1 \mathbf{g}(\Theta_s)^\top \delta\Theta ds \tag{21}$$

$$= -\sum_{i \in \mathcal{S}} \theta_i^\tau \int_0^1 \mathbf{g}_i(\Theta_s)ds. \tag{22}$$

The loss difference is written as a path integral and, using the chain rule with the constant velocity $\delta\Theta = -\theta^\tau$, reduces to a weighted path-average of the removed gradient components. This compact form directly quantifies each deleted parameter's marginal contribution to $\Delta\mathcal{L}(\mathcal{S})$.

### 3. Expanding the *path-wise* gradient

$$\mathbf{g}(\Theta_s) = \mathbf{g}^\tau + \int_0^s \mathbf{H}(\Theta_t)\delta\Theta dt, \tag{23}$$

$$\mathbf{g}_i(\Theta_s) = \mathbf{g}_i^\tau - \sum_{j \in \mathcal{S}} \theta_j^\tau \int_0^s \mathbf{H}_{ij}(\Theta_t)dt, \tag{24}$$

$$\Delta\mathcal{L}(\mathcal{S}) = -\sum_{i \in \mathcal{S}} \mathbf{g}_i^\tau \theta_i^\tau + \sum_{i,j \in \mathcal{S}} \theta_i^\tau \theta_j^\tau \int_0^1 \int_0^s \mathbf{H}_{ij}(\Theta_t)dt\,ds. \tag{25}$$

Integrating the Hessian along the optimisation path yields the path-dependent gradient (first two lines), and substituting this result into the loss variation condenses the double integral, showing how curvature accumulates through the $\mathbf{H}_{ij}$ "path-area" term.

By exchanging the order of integration, the nested double integral collapses to the single-integral kernel $(1-t)$, whose substitution into the path-dependent loss expansion yields the quadratic form below.

$$\int_0^1 \int_0^s (\cdot)dt\,ds = \int_0^1 \int_t^1 (\cdot)ds\,dt = \int_0^1 (1-t)(\cdot)dt, \tag{26}$$

$$\Delta\mathcal{L}(\mathcal{S}) = -\sum_{i \in \mathcal{S}} \mathbf{g}_i^\tau \theta_i^\tau + \sum_{i,j \in \mathcal{S}} \theta_i^\tau \left[\int_0^1 (1-t)\mathbf{H}_{ij}(\Theta_t)dt\right]\theta_j^\tau. \tag{27}$$

## 4. Defining the $\mathbf{w}_{ij}^{(\mathcal{S})}$

Defining the dimensionless weight $\mathbf{w}_{ij}^{(\mathcal{S})}$ as the path-averaged curvature normalised by the terminal Hessian allows eq. (29) to be recast in a compact quadratic form where all trajectory dependence is absorbed into $\mathbf{w}_{ij}^{(\mathcal{S})}$.

$$\mathbf{w}_{ij}^{(\mathcal{S})} := \frac{1}{\mathbf{H}_{ij}^\tau} \int_0^1 (1-t)\mathbf{H}_{ij}(\Theta_t)dt, \tag{28}$$

$$\Delta\mathcal{L}(\mathcal{S}) \approx -\sum_{i\in\mathcal{S}} \mathbf{g}_i^\tau \theta_i^\tau + \frac{1}{2} \sum_{i,j\in\mathcal{S}} \mathbf{w}_{ij}^{(\mathcal{S})} \mathbf{H}_{ij}^\tau \theta_i^\tau \theta_j^\tau. \tag{29}$$

**Why this definition?**

(i) The factor $(1-t)$ is the exact kernel that emerges when the double integral in eq (28) is reordered; it ensures the "$\frac{1}{2}$" symmetry familiar from second-order Taylor series.

(ii) Dividing by $\mathbf{H}_{ij}^\tau$ *normalises* the path-average so that $\mathbf{w}_{ij}^{(\mathcal{S})}$ is a pure number measuring how much the curvature between coordinates $(i,j)$ varies *relative to its value at* $\Theta^\tau$. In particular, $\mathbf{w}_{ij}^{(\mathcal{S})} = 1$ if $\mathbf{H}_{ij}$ is constant along the path.

## 5. Limiting behaviour as the removed weights vanish

Suppose $\|\theta_\mathcal{S}\| \to 0$. Then $\|\delta\Theta\| \to 0$, so the entire path collapses into an arbitrarily small neighbourhood of $\Theta^\tau$. By continuity of the Hessian, $\mathbf{H}_{ij}(\Theta_t) \to \mathbf{H}_{ij}^\tau$ uniformly, and

$$\mathbf{w}_{ij}^{(\mathcal{S})} = 1 + \mathcal{O}(\|\theta_\mathcal{S}\|). \tag{30}$$

Consequently eq (29) reduces smoothly to the usual second-order Taylor estimate

$$\Delta\mathcal{L}(\mathcal{S}) \approx -\sum_{i\in\mathcal{S}} \mathbf{g}_i^\tau \theta_i^\tau + \frac{1}{2} \sum_{i,j\in\mathcal{S}} \mathbf{H}_{ij}^\tau \theta_i^\tau \theta_j^\tau. \tag{31}$$

This completes the explicit path-integral proof, while clarifying the origin and interpretation of the weighting coefficients $\mathbf{w}_{ij}^{(\mathcal{S})}$.

### C.3 Combine Marginal Utility Function

In this appendix, we provide the detailed derivation of the closed-form Shapley value approximation given in equation 7, now incorporating path-based curvature weights.

**Setup.** Recall that the Shapley value for a parameter $\theta_i$ under a utility function $U(\Theta) = -\mathcal{L}(\Theta)$ is defined as:

$$\phi_i(U) = \frac{1}{M} \sum_{k=1}^M \binom{M-1}{k-1}^{-1} \sum_{\substack{S\subseteq\Theta\setminus\{i\} \\ |S|=k-1}} [U(S\cup\{i\}) - U(S)], \tag{32}$$

where $M = |\Theta|$ is the total number of parameters.

**Marginal contribution approximation.** Based on the path-integrated second-order approximation, the marginal utility of adding parameter $\theta_i$ to subset $S$ is:

$$U(S\cup\{i\}) - U(S) \approx -\mathbf{g}_i^\tau \theta_i - \frac{1}{2}\mathbf{w}_{ii}^{(i)}(\theta_i)^2 \mathbf{H}_{ii}^\tau - \theta_i \sum_{j\in S} \mathbf{w}_{ij}^{(i)} \mathbf{H}_{ij}^\tau \theta_j, \tag{33}$$

where $\mathbf{w}_{ij}^{(i)} \in [0,1]$ denotes a curvature-weighted average over the interpolation path involving parameters $i$ and $j$.

**Averaging over subsets.** Substituting equation 33 into equation 32, we have:

$$\phi_i(U) = \frac{1}{M}\sum_{k=1}^{M}\binom{M-1}{k-1}^{-1}\sum_{\substack{S\subseteq\Theta\setminus\{i\}\\|S|=k-1}}\left(-\mathbf{g}_i^\tau\theta_i - \frac{1}{2}\mathbf{w}_{ii}^{(i)}\theta_i^2\mathbf{H}_{ii}^\tau - \theta_i\sum_{j\in S}\mathbf{w}_{ij}^{(i)}\mathbf{H}_{ij}^\tau\theta_j\right)$$

$$= \left(-\mathbf{g}_i^\tau\theta_i - \frac{1}{2}\mathbf{w}_{ii}^{(i)}\theta_i^2\mathbf{H}_{ii}^\tau\right) - \frac{1}{M}\theta_i\sum_{k=1}^{M}\binom{M-1}{k-1}^{-1}\sum_{\substack{S\subseteq\Theta\setminus\{i\}\\|S|=k-1}}\sum_{j\in S}\mathbf{w}_{ij}^{(i)}\mathbf{H}_{ij}^\tau\theta_j. \qquad (34)$$

**Simplifying the interaction term.** Fixing index $j\neq i$, the number of subsets $S$ of size $k-1$ that contain $j$ is $\binom{M-2}{k-2}$, so:

$$\sum_{\substack{S\subseteq\Theta\setminus\{i\}\\|S|=k-1}}\sum_{j\in S}\mathbf{w}_{ij}^{(i)}\mathbf{H}_{ij}^\tau\theta_j = \sum_{j\neq i}\binom{M-2}{k-2}\mathbf{w}_{ij}^{(i)}\mathbf{H}_{ij}^\tau\theta_j. \qquad (35)$$

Substituting back:

$$\phi_i(U) = \left(-\mathbf{g}_i^\tau\theta_i - \frac{1}{2}\mathbf{w}_{ii}^{(i)}\theta_i^2\mathbf{H}_{ii}^\tau\right) - \frac{1}{M}\theta_i\sum_{j\neq i}\mathbf{w}_{ij}^{(i)}\mathbf{H}_{ij}^\tau\theta_j\sum_{k=1}^{M}\frac{\binom{M-2}{k-2}}{\binom{M-1}{k-1}}. \qquad (36)$$

**Evaluating the combinatorial ratio.** As before,

$$\frac{\binom{M-2}{k-2}}{\binom{M-1}{k-1}} = \frac{k-1}{M-1}, \quad \sum_{k=1}^{M}\frac{k-1}{M-1} = \frac{M}{2}. \qquad (37)$$

**Final closed-form expression.** Substituting the result:

$$\phi_i(U) = -\mathbf{g}_i^\tau\theta_i - \frac{1}{2}\mathbf{w}_{ii}^{(i)}\theta_i^2\mathbf{H}_{ii}^\tau - \frac{1}{2}\theta_i\sum_{j\neq i}\mathbf{w}_{ij}^{(i)}\mathbf{H}_{ij}^\tau\theta_j. \qquad (38)$$

This generalizes the original expression in equation 7 by introducing path-weighted curvature terms $\mathbf{w}_{ij}^{(i)}$, derived from integrating along removal trajectories. $\qquad\square$

## C.4 Taylor Expansion Remainder Bound for Shapley Value Approximation

Our Shapley value approximation in §4.1 is derived from a path integral formulation, which computes the expected marginal contribution of each parameter along a continuous interpolation path. When $\theta_i \to 0$, this integral can be well-approximated by the second-order Taylor expansion of the utility function around the current parameter vector $\Theta^\tau$. This appendix quantifies the approximation error by deriving a uniform bound on the remainder term in the Taylor expansion.

**Second-order expansion with remainder.** For any parameter vector $\theta$ in a neighborhood of a reference point $\theta^*$,

$$\mathcal{L}(\theta) = \mathcal{L}(\theta^*) + \nabla\mathcal{L}(\theta^*)^\top(\theta-\theta^*) + \tfrac{1}{2}(\theta-\theta^*)^\top\nabla^2\mathcal{L}(\theta^*)(\theta-\theta^*) + \mathcal{R}_3(\theta,\theta^*), \qquad (39)$$

where $\mathcal{R}_3$ collects all third- and higher-order terms. By Taylor's theorem there exists $t\in(0,1)$ such that

$$|\mathcal{R}_3(\theta,\theta^*)| \leq \frac{1}{6}\|\theta-\theta^*\|^3\big\|\nabla^3\mathcal{L}\big(\theta^*+t(\theta-\theta^*)\big)\big\|. \qquad (40)$$

**Propagation to the Shapley estimate.** Denote by $\phi_i^\star$ the *true* Shapley value and by $\tilde{\phi}_i$ the second-order approximation given in Eq. (7). For each coalition $\mathcal{S}\subseteq\Theta\setminus\{\theta_i\}$ we define the Taylor error

$$\varepsilon_i(\mathcal{S}) = \left|\mathcal{R}_3(\Theta^{\tau,-\mathcal{S}\cup\{i\}},\Theta^\tau) - \mathcal{R}_3(\Theta^{\tau,-\mathcal{S}},\Theta^\tau)\right|. \qquad (41)$$

Using the linearity of the Shapley aggregator we have

$$\left|\phi_i^\star - \tilde{\phi}_i\right| \leq \frac{1}{M}\sum_{\mathcal{S}\subseteq\Theta\setminus\{i\}}w_{\mathcal{S}}\cdot\varepsilon_i(\mathcal{S}), \quad \text{where} \quad w_{\mathcal{S}} = \binom{M-1}{|\mathcal{S}|}^{-1}. \qquad (42)$$

**Uniform bound under smoothness and small perturbation.** Assume the following: (1) The third derivative of the loss is uniformly bounded within a ball of radius $\varepsilon$ around $\Theta^\tau$, i.e., $\|\nabla^3 \mathcal{L}(\theta)\| \leq C$ for all $\|\theta - \Theta^\tau\| \leq \varepsilon$; (2) Each single-parameter removal satisfies $\|\Theta^\tau - \Theta^{\tau,-j}\| \leq \varepsilon$. This holds under the "set-to-zero" removal rule, with $\varepsilon = \max_j |\theta_j^\tau|$. Under these assumptions, Eq. (40) implies that $\varepsilon_i(\mathcal{S}) \leq \frac{C}{6}\varepsilon^3$ for every subset $\mathcal{S}$, and the summation in Eq. (42) becomes:

$$\left| \phi_i^\star - \tilde{\phi}_i \right| \leq \frac{C}{6}\varepsilon^3. \tag{43}$$

The cubic dependence on $\varepsilon$ implies that the approximation error rapidly vanishes as training converges and parameter updates shrink. The constant $C$ reflects the worst-case third-order curvature in the local neighborhood. In practice, we observe that this bound is small for modern architectures, which supports the empirical accuracy of our closed-form estimate.

### C.5 Proof of Shapley Axiom Properties under Second-order Approximation

We provide formal justifications that the closed-form parameter-level Shapley expression equation 7 approximately satisfies the four core Shapley axioms under second-order Taylor approximation.

#### C.5.1 Null Player

**Claim:** If a parameter $\theta_i$ has no effect on any subset of parameters, then $\phi_i = 0$.

**Proof:** From equation 7, we have:

$$\phi_i = -\mathbf{g}_i^\tau \theta_i^\tau - \tfrac{1}{2}\mathbf{w}_{ii}^{(i)}(\theta_i^\tau)^2 \mathbf{H}_{ii}^\tau - \theta_i^\tau \sum_{j \in \mathcal{S}} \mathbf{w}_{ij}^{(\mathcal{S})} \mathbf{H}_{ij}^\tau \theta_j^\tau. \tag{44}$$

If $\mathbf{g}_i^\tau = 0$ and $\mathbf{H}_{ij}^\tau = 0$ for all $j$, and $\theta_i = 0$ or finite, then all terms vanish and $\phi_i = 0$. This corresponds exactly to the definition of a null player in Shapley theory.

$\square$

#### C.5.2 Symmetry

**Claim:** If parameters $\theta_i$ and $\theta_j$ contribute identically to the loss and interact identically with all other parameters, then $\phi_i = \phi_j$.

**Proof:** Assume the following: - $\theta_i = \theta_j$, - $\mathbf{g}_i^\tau = \mathbf{g}_j^\tau$, - $\mathbf{H}_{ii}^\tau = \mathbf{H}_{jj}^\tau$, - $\mathbf{H}_{ik}^\tau = \mathbf{H}_{jk}^\tau$ for all $k \neq i, j$, - $\mathbf{H}_{ki}^\tau = \mathbf{H}_{kj}^\tau$ (by symmetry of Hessian).

Then by substituting into equation 7, we have:

$$\phi_i = -\mathbf{g}_i^\tau \theta_i^\tau - \tfrac{1}{2}\mathbf{w}_{ii}^{(i)}(\theta_i^\tau)^2 \mathbf{H}_{ii}^\tau - \theta_i^\tau \sum_{k \in \mathcal{S}} \mathbf{w}_{ik}^{(\mathcal{S})} \mathbf{H}_{ik}^\tau \theta_k^\tau. \tag{45}$$

$$\phi_j = -\mathbf{g}_j^\tau \theta_j^\tau - \tfrac{1}{2}\mathbf{w}_{jj}^{(j)}(\theta_j^\tau)^2 \mathbf{H}_{jj}^\tau - \theta_j^\tau \sum_{k \in \mathcal{S}} \mathbf{w}_{jk}^{(\mathcal{S})} \mathbf{H}_{ik}^\tau \theta_k^\tau. \tag{46}$$

Since all terms are identical under the above symmetry assumptions, $\phi_i = \phi_j$.

$\square$

#### C.5.3 Linearity

**Claim:** If utility functions are linearly combined as $U = \alpha U_1 + \beta U_2$, then the Shapley values satisfy $\phi_i(U) = \alpha \phi_i(U_1) + \beta \phi_i(U_2)$.

**Proof:** Let the losses be $\mathcal{L}_1, \mathcal{L}_2$, and define:

$$\mathcal{L} = \alpha\mathcal{L}_1 + \beta\mathcal{L}_2, \quad U = -\mathcal{L},$$

$$\mathbf{g}_i = \partial_{\theta_i}\mathcal{L} = \alpha\mathbf{g}_i^{(1)} + \beta\mathbf{g}_i^{(2)}, \tag{47}$$

$$\mathbf{H}_{ij} = \partial_{\theta_j}\partial_{\theta_i}\mathcal{L} = \alpha\mathbf{H}_{ij}^{(1)} + \beta\mathbf{H}_{ij}^{(2)}.$$

Substituting these expressions into the closed-form formula for $\phi_i$ gives

$$\phi_i = \alpha\phi_i^{(1)} + \beta\phi_i^{(2)}, \tag{48}$$

thus confirming exact linearity of the contribution with respect to the loss weights.

$\square$

### C.5.4 Efficiency (Approximate)

**Claim.** *The total Shapley value approximates the total utility:*

$$\sum_{i=1}^{M}\phi_i \approx U(\Theta) - U(\emptyset). \tag{49}$$

**Proof.**

$$\tilde{\mathbf{g}}_i := \alpha\mathbf{g}_i^{(1)} + \beta\mathbf{g}_i^{(2)}, \tag{50}$$

$$\tilde{\mathbf{H}}_{ii} := \alpha\mathbf{w}_{ii}^{(1)}\mathbf{H}_{ii}^{(1)} + \beta\mathbf{w}_{ii}^{(2)}\mathbf{H}_{ii}^{(2)}, \tag{51}$$

$$\tilde{\mathbf{H}}_{ij} := \alpha\mathbf{w}_{ij}^{(\mathcal{S},1)}\mathbf{H}_{ij}^{(1)} + \beta\mathbf{w}_{ij}^{(\mathcal{S},2)}\mathbf{H}_{ij}^{(2)}, \quad (i \neq j). \tag{52}$$

*These weighted quantities merge the task-specific gradients and curvatures into a single set of effective statistics.*

$$\phi_i = -\tilde{\mathbf{g}}_i\theta_i - \tfrac{1}{2}\tilde{\mathbf{H}}_{ii}\theta_i^2 - \tfrac{1}{2}\theta_i\sum_{j\neq i}\tilde{\mathbf{H}}_{ij}\theta_j. \tag{53}$$

*With the aggregates in hand, each Shapley value becomes a concise quadratic contribution to loss reduction.*

$$\Delta\mathcal{L} := \mathcal{L}(\Theta) - \mathcal{L}(0) \approx \tilde{\mathbf{g}}^{\top}\boldsymbol{\theta} + \tfrac{1}{2}\boldsymbol{\theta}^{\top}\tilde{\mathbf{H}}\boldsymbol{\theta} = \sum_i\tilde{\mathbf{g}}_i\theta_i + \tfrac{1}{2}\sum_{i,j}\theta_i\tilde{\mathbf{H}}_{ij}\theta_j. \tag{54}$$

A second-order Taylor expansion around $\Theta = 0$ yields an identical quadratic form in the same aggregated terms.

$$\sum_{i=1}^{M}\phi_i = -\sum_i\tilde{\mathbf{g}}_i\theta_i - \tfrac{1}{2}\sum_{i,j}\theta_i\tilde{\mathbf{H}}_{ij}\theta_j \approx -\Delta\mathcal{L}. \tag{55}$$

Summing the individual payoffs therefore reproduces the negative of the Taylor approximation, up to cubic errors.

$$\sum_{i=1}^{M}\phi_i \approx U(\Theta) - U(\emptyset). \tag{56}$$

Since $U = -\mathcal{L}$, this matches the utility gain and thus verifies Shapley efficiency to second order. $\square$

## D  Fisher Information Matrix as an Approximation to the Hessian

In this section, we explain why the **Fisher Information Matrix (FIM)** is often used as a tractable approximation to the Hessian of the loss function, particularly in settings where the loss arises from the negative log-likelihood.

**Log-likelihood and the Score Function**   Consider a probabilistic model $p(x \mid \theta)$ (i.e. LLMs) with parameter vector $\theta \in \mathbb{R}^d$, and let $x$ denote a single observed data point. The *log-likelihood* function is given by $\log p(x \mid \theta)$, and its gradient with respect to $\theta$ is known as the **score function**: $s(\theta) := \nabla_\theta \log p(x \mid \theta)$. The score function measures the sensitivity of the log-likelihood to changes in the parameters. Under standard regularity conditions (e.g., differentiability, integrability, and dominated convergence), the expectation of the score vanishes: $\mathbb{E}_{x \sim p(x|\theta)}[s(\theta)] = 0$.

**Definition and Interpretation of the Fisher Matrix**   The **Fisher Information Matrix** is defined as the covariance (second moment) of the score function: $F(\theta) := \mathbb{E}_{x \sim p(x|\theta)}\left[s(\theta)s(\theta)^\top\right]$. Intuitively, it quantifies how much information the data carries about the parameters: if small changes in $\theta$ cause large variations in the likelihood, then the entries of $F(\theta)$ will be large.

**Connection to the Hessian of the Log-Likelihood**   To relate this to the Hessian, consider the second derivative (Hessian) of the log-likelihood: $H_{\mathrm{loglik}}(\theta) := \nabla_\theta^2 \log p(x \mid \theta)$. Using the identity $\nabla_\theta^2 \log p(x \mid \theta) = \frac{\nabla_\theta^2 p(x|\theta)}{p(x|\theta)} - s(\theta)s(\theta)^\top$, and the fact that the total probability integrates to one (i.e., $\int p(x \mid \theta) dx = 1$), we obtain:

$$\mathbb{E}_{x \sim p(x|\theta)}\left[\nabla_\theta^2 \log p(x \mid \theta)\right] = \nabla_\theta^2 \int p(x \mid \theta) dx - \mathbb{E}\left[s(\theta)s(\theta)^\top\right] = \nabla_\theta^2(1) - F(\theta) = -F(\theta). \quad (57)$$

This is a well-known result in statistics:

$$F(\theta) = -\mathbb{E}_{x \sim p(x|\theta)}\left[\nabla_\theta^2 \log p(x \mid \theta)\right]. \quad (58)$$

That is, the Fisher Information Matrix equals the negative expected Hessian of the log-likelihood.

**From Log-Likelihood to Loss Function**   In machine learning and statistical estimation, it is common to define the loss function as the negative log-likelihood: $\mathcal{L}(\theta) := -\log p(x \mid \theta)$, whose Hessian becomes

$$H_\mathcal{L}(\theta) := \nabla_\theta^2 \mathcal{L}(\theta) = -\nabla_\theta^2 \log p(x \mid \theta). \quad (59)$$

Taking expectations yields:

$$\mathbb{E}_{x \sim p(x|\theta)}\left[\nabla_\theta^2 \mathcal{L}(\theta)\right] = F(\theta), \quad (60)$$

demonstrating that the Fisher matrix equals the expected curvature of the loss landscape.

**Empirical Fisher Matrix and Practical Considerations**   In practice, the true data distribution $p(x \mid \theta)$ is unknown, and we instead work with a finite dataset $\{x_i\}_{i=1}^N$. The **empirical Fisher matrix** is computed as:

$$\widehat{F}(\theta) := \frac{1}{N} \sum_{i=1}^N (\nabla_\theta \log p(x_i \mid \theta))(\nabla_\theta \log p(x_i \mid \theta))^\top. \quad (61)$$

This approximation has several advantages: it avoids computing second derivatives, is simple to implement, and is compatible with stochastic optimization techniques such as mini-batch gradient descent.

# E   Applications of MODEL SHAPLEY

Leveraging the estimated Shapley values $\{\phi_i\}$, **MODEL SHAPLEY** facilitates a variety of downstream applications centered on functional attribution at the parameter level. These applications span from interpreting the internal mechanisms of the model, to actively manipulating its behavior during inference, fine-tuning critical components, and guiding parameter-efficient model quantization.

In the following, we illustrate how **MODEL SHAPLEY** enables parameter-level functional analysis through four key stages: ❶ constructing a task-specific dataset for attribution, ❷ determining the granularity of analysis (e.g., neuron, layer, or attention head), ❸ applying Shapley-based attribution to identify important units, and ❹ utilizing the results in downstream interventions.

## E.1 Step 1: Constructing the Probe Dataset

To attribute functionality at the parameter level, we first construct a probe dataset $\mathcal{D}_{\mathcal{F}} = \{(x_i, y_i)\}_{i=1}^N$ tailored to a specific objective $\mathcal{F}$ (e.g., robustness, factual recall, reasoning). Unlike training or evaluation datasets, this probe dataset is explicitly designed to activate the model's behavior with respect to the target capability, thereby enabling meaningful attribution.

The structure of $\mathcal{D}_{\mathcal{F}}$ depends on the task. For instance, in retrieval-augmented generation (RAG), each input $x_i$ consists of a {query, retrieved document} pair, and the output $y_i$ is the corresponding response generated by the model. In code generation tasks, $x_i$ may be a natural language prompt or partial program, with $y_i$ being the expected code completion. For robustness assessment, $x_i$ could be adversarially perturbed inputs, while $y_i$ reflects the desired stable output under perturbation. Across these diverse settings, the probe dataset serves as a lens through which task-specific model behavior is revealed.

By computing Shapley values over $\mathcal{D}_{\mathcal{F}}$, we estimate each parameter's marginal contribution to the model's ability to satisfy objective $\mathcal{F}$. This attribution highlights the most functionally sensitive units—those whose perturbation or removal would most affect the model's performance on the given task.

## E.2 Step 2: Defining Attribution Granularity

To enable interpretable functional attribution, we first define the level of granularity at which model parameters are examined. Transformer architectures are inherently modular, composed of repeated layers that include Multi-Head Attention (MHA) mechanisms and Feed-Forward Networks (FFNs). This modularity naturally gives rise to three commonly used units of attribution: ***neurons***, ***attention heads***, and ***layers***. Each level captures different aspects of model behavior and presents trade-offs between interpretability, attribution resolution, and practical utility for intervention.

**Neuron-Level Attribution.** A neuron refers to an individual hidden unit within the FFN of a given Transformer layer. Specifically, the $j$-th neuron in the $i$-th FFN layer computes:

$$\mathbf{neuron}_{i,j} = \text{act\_fn}\Big( [\tilde{\mathbf{h}}^i \mathbf{W}_1^i + \mathbf{b}_1^i]_j \Big), \tag{62}$$

where $\tilde{\mathbf{h}}^i \in \mathbb{R}^d$ is the output representation from the MHA submodule in layer $i$, $\mathbf{W}_1^i \in \mathbb{R}^{d \times 4d}$ and $\mathbf{b}_1^i \in \mathbb{R}^{4d}$ are the weight and bias parameters of the FFN input projection, and $\text{act\_fn}(\cdot)$ is typically a GELU or SiLU nonlinearity. A neuron is considered *active* if its post-activation value is nonzero. Attribution at this granularity enables highly localized functional analysis, though it may be sensitive to activation noise and task-specific variance.

**Attention Head-Level Attribution.** Each MHA module consists of multiple heads, each responsible for computing an independent attention pattern. The output of the $h$-th head in the $i$-th layer is given by:

$$\mathbf{head}_h^i = \text{Attn}\big( \mathbf{h}^{i-1} \mathbf{W}_{q,h}^i, \mathbf{H}^{i-1} \mathbf{W}_{k,h}^i, \mathbf{H}^{i-1} \mathbf{W}_{v,h}^i \big), \tag{63}$$

where $\mathbf{h}^{i-1} \in \mathbb{R}^d$ denotes the representation of a single token from the previous layer, $\mathbf{H}^{i-1} \in \mathbb{R}^{L \times d}$ denotes the sequence of hidden states, and $\mathbf{W}_{q,h}^i, \mathbf{W}_{k,h}^i, \mathbf{W}_{v,h}^i \in \mathbb{R}^{d \times d/H}$ are the learned projection matrices for queries, keys, and values, respectively. Attention heads often specialize in capturing distinct structural or linguistic features (e.g., syntax, coreference, position), making this level of attribution particularly effective for interpreting distributed attention behavior and its contribution to model predictions.

**Layer-Level Attribution.** Each Transformer layer comprises the full computation block including all attention heads and the FFN. The forward pass through layer $i$ can be expressed as:

$$\tilde{\mathbf{h}}^i = \text{Concat}(\mathbf{head}_1^i, ..., \mathbf{head}_H^i) \mathbf{W}_o^i + \mathbf{b}_o^i, \quad \mathbf{h}^i = \text{act\_fn}(\tilde{\mathbf{h}}^i \mathbf{W}_1^i + \mathbf{b}_1^i) \mathbf{W}_2^i + \mathbf{b}_2^i, \tag{64}$$

where $\mathbf{W}_o^i \in \mathbb{R}^{d \times d}$ and $\mathbf{b}_o^i \in \mathbb{R}^d$ are the output projection weights and biases for the attention module, and $\mathbf{W}_1^i, \mathbf{W}_2^i$ are the FFN projection matrices with corresponding biases $\mathbf{b}_1^i, \mathbf{b}_2^i$. Attribution at the layer level provides a coarser but more robust perspective on where critical computations occur, and is often suitable for interventions such as pruning, freezing, or targeted parameterization.

### E.3 Step 3: Shapley-Based Attribution of Parameter Importance

To quantify the task-specific importance of individual parameter units, we adopt a Shapley value framework grounded in cooperative game theory. Shapley values offer a principled approach to measuring the marginal utility of each parameter, averaged over all possible subsets of cooperating units. This provides a theoretically justified metric for attributing functional relevance.

However, exact Shapley computation is intractable in large-scale neural networks due to the exponential number of subsets. To address this, we derive a closed-form approximation based on a second-order Taylor expansion of the loss function $\mathcal{L}(\Theta)$. Under the common assumption that removing a parameter can be approximated by zeroing it out, the Shapley value $\phi_i$ for parameter $\theta_i$ is given by:

$$\phi_i = \underbrace{-\mathbf{g}_i^\tau \theta_i}_{\text{(1) Individual Importance}} \underbrace{-\frac{1}{2}\mathbf{w}_{ii}^{(i)}\theta_i^2 \mathbf{H}_{ii}^\tau - \frac{1}{2}\theta_i \sum_{j \neq i} \mathbf{w}_{ij}^{(\mathcal{S})} \mathbf{H}_{ij}^\tau \theta_j}_{\text{(2) Cooperative Interactions}}, \tag{65}$$

where $\mathbf{g}_i = \frac{\partial \mathcal{L}}{\partial \theta_i}$, and $\mathbf{H}_{ij}$ denotes entries of the Hessian. The first term captures the individual salience of the parameter, while the remaining terms account for higher-order cooperative interactions.

To scale this computation in practice, we employ several approximations:

- Gradients and Hessian-vector products are estimated on mini-batches, enabling per-step online approximation of $\phi_i$.
- We stabilize noisy estimates with an exponential moving average (EWMA): $\widehat{\phi}_i^\tau = (1-\alpha)\widehat{\phi}_i^{\tau-1} + \alpha\phi_i^\tau$.
- The Hessian is approximated using the empirical Fisher Information Matrix (FIM): $\mathbf{F}(\Theta) := \mathbb{E}_{(x,y)}\left[\nabla_\Theta \mathcal{L}(\Theta;x,y)\nabla_\Theta \mathcal{L}(\Theta;x,y)^\top\right]$, which is more stable and tractable under stochastic training dynamics.
- To reduce memory and computational overhead, we apply a **blockwise approximation**, aggregating importance scores within structured units (e.g., neurons in a layer or heads in an MHA block) using local statistics.

After computing $\phi_i$ for all parameter units, we identify the top-$\alpha\%$ most important units by ranking their Shapley values. For coarser-grained structures such as attention heads or entire layers, we perform **mean pooling over neuron-level scores** within each unit:

$$\phi_{\text{head}}^{(i)} = \frac{1}{|\Theta_{\text{head}}^{(i)}|} \sum_{\theta_j \in \Theta_{\text{head}}^{(i)}} \phi_j, \quad \phi_{\text{layer}}^{(i)} = \frac{1}{|\Theta_{\text{layer}}^{(i)}|} \sum_{\theta_j \in \Theta_{\text{layer}}^{(i)}} \phi_j. \tag{66}$$

This pooling ensures that higher-level Shapley attribution remains grounded in fine-grained parameter sensitivity, while enabling consistent scoring across different levels of model granularity. grained architectural intervention.

### E.4 Step 4: Applications of Attributed Parameter Units

The estimated Shapley scores $\{\phi_i\}$ identify parameter units that are functionally critical for a given task. These scores support a variety of downstream applications aimed at interpreting model behavior, reinforcing specific capabilities, or guiding efficient intervention. We describe two representative use cases below.

#### E.4.1 Direct Usage for Functional Probing and Activation Augmentation

Without any further training, the activations of high-importance parameter units can be used to probe internal model representations or to modulate inference behavior. Let $\mathcal{I}_\alpha = \{i \mid \phi_i \text{ ranks in top-}\alpha\%\}$ denote the set of top-scoring parameters.

At inference time, we extract the activations $a_i(x)$ of the important units and use them as feature inputs to a probing model:

$$\mathcal{F}_{\text{probe}}(x) = [a_i(x)]_{i \in \mathcal{I}_\alpha}, \tag{67}$$

where $\mathcal{F}_{\text{probe}}$ is a downstream classifier trained to detect specific knowledge, behaviors, or internal states. This enables lightweight interpretability tools for factual recall, knowledge localization, fairness analysis, or safety probing.

Alternatively, we can modulate the model's output by augmenting the activation of important units:

$$a_i^{\text{enhanced}}(x) = a_i(x) + \Delta \cdot \mathbf{1}[i \in \mathcal{I}_\alpha], \tag{68}$$

where $\Delta$ is a tunable scalar offset. This form of activation intervention parallels the concept of functional or concept vectors [41, 39], and can be used to amplify model behaviors associated with certain capabilities.

### E.4.2 Targeted Fine-Tuning for Capability Enhancement

Rather than updating the entire model, we can focus learning on the most functionally relevant parameter units identified via $\{\phi_i\}$. During fine-tuning, we restrict gradient updates to only the top-ranked parameters $i \in \mathcal{I}_\alpha$, while freezing the rest:

$$\theta_i \leftarrow \theta_i - \eta \cdot \nabla_{\theta_i} \mathcal{L}, \quad \text{for } i \in \mathcal{I}_\alpha, \tag{69}$$

$$\theta_j \leftarrow \theta_j, \quad \text{for } j \notin \mathcal{I}_\alpha, \tag{70}$$

where $\eta$ is the learning rate. This strategy allows for parameter-efficient fine-tuning by concentrating updates on task-relevant components while reducing overfitting and preserving prior capabilities. Such targeted adaptation is particularly beneficial in continual learning, domain adaptation, or few-shot regimes.

### E.4.3 Shapley-Guided Quantization

OBD evaluates importance through the first-order product $|g_i \theta_i|$. OBS and GPTQ extend this idea by incorporating non-diagonal Hessian entries so that parameter interactions are partially compensated.

In a network that has already converged, the first-order gradient term $\partial \mathcal{L} / \partial \theta_i$ is typically close to zero (see Eq. (7)), so the loss change $\Delta \mathcal{L}(\theta_i)$ is dominated by the second-order component

$$\phi_i^{(2)} = -\tfrac{1}{2}\theta_i^2 \mathbf{H}_{ii} - \tfrac{1}{2}\theta_i \sum_{j \neq i} \mathbf{H}_{ij}\theta_j.$$

The cross term $\sum_{j \neq i} \mathbf{H}_{ij}\theta_j$ is already handled by GPTQ's inverse update. What remains unaccounted for is the task-specific magnitude of the diagonal piece $-\tfrac{1}{2}\theta_i^2 \mathbf{H}_{ii}$. We therefore correct $\mathbf{H}_{ii}$ itself rather than modifying the update order.

Let $\phi_i^{(2)}$ denote the empirical second-order Shapley estimate obtained with the Fisher-block approximation of Section 4.3. Define the per-parameter correction factor and modified diagonal entry

$$w_i := \left| \frac{\phi_i^{(2)}}{\tfrac{1}{2}\theta_i^2 \mathbf{H}_{ii} + \varepsilon} \right|, \qquad \tilde{\mathbf{H}}_{ii} \leftarrow w_i \beta \mathbf{H}_{ii} + (1-\beta)\mathbf{H}_{ii}, \tag{71}$$

where $\varepsilon > 0$ ensures numerical stability and $\beta \in (0,1]$ controls the strength of the Shapley injection. If $w_i > 1$ the diagonal term is boosted; if $w_i < 1$ it is attenuated.

Replacing $\mathbf{H}_{ii}$ by $\tilde{\mathbf{H}}_{ii}$ leaves the Woodbury-style inverse update in Algorithm E.1 unchanged, so the overall $\mathcal{O}(dk^2)$ complexity is identical to vanilla GPTQ. The convergence proof in Appendix B.4 remains valid after substituting $\mathbf{H} \mapsto \tilde{\mathbf{H}}$.

Equation equation 71 follows directly from the efficiency axiom applied to the second-order expansion in Theorem 4.6. By isolating the quadratic diagonal term inside $\phi_i^{(2)}$, we obtain a scale-invariant ratio that indicates whether the naive quadratic estimate under- or over-explains the cooperative loss increase attributed to weight $i$.

OBD relies on gradients, OBS and GPTQ already harness off-diagonal curvature, and our Shapley-guided diagonal correction completes the picture by aligning each diagonal Hessian entry with its cooperative, task-specific contribution. The resulting $\tilde{\mathbf{H}}$ integrates seamlessly into GPTQ, yielding stronger accuracy–compression trade-offs without sacrificing speed.

The overall procedure is outlined in **Algorithm** 1.

---

**Algorithm 1** Shapley-Guided Quantization with Corrected Hessian

---

**Require:** Weight matrix $\mathbf{W}$, inverse Hessian $\mathbf{H}^{-1}$, Shapley scores $\{\phi_i^{(2)}\}$, block size $B$, reweight parameter $\beta \in [0,1]$
**Ensure:** Quantized weight matrix $\mathbf{Q}$

1: Initialize $\mathbf{Q} \leftarrow \mathbf{0}$, $\mathbf{E} \leftarrow \mathbf{0}$
2: **for** $i = 0$ to $N$ in steps of $B$ **do**                                      ▷ Iterate over blocks
3:     **for** $j = i$ to $i+B-1$ **do**
4:         $s_j \leftarrow \frac{1}{2} \cdot \theta_j^2 \cdot H_{jj}$                           ▷ Estimate standard Hessian contribution
5:         $w_j \leftarrow \left| \phi_j^{(2)} / (s_j + \varepsilon) \right|$                     ▷ Compute Shapley-based correction weight
6:         $\tilde{H}_{jj} \leftarrow w_j \cdot \beta H_{jj} + (1-\beta) \cdot H_{jj}$                    ▷ Rescale diagonal Hessian entry
7:         $\mathbf{Q}_{:,j} \leftarrow \mathrm{quant}(\mathbf{W}_{:,j})$                        ▷ Quantize column $j$
8:         $\mathbf{E}_{:,j-i} \leftarrow (\mathbf{W}_{:,j} - \mathbf{Q}_{:,j}) / \tilde{H}_{jj}$                 ▷ Estimate quantization error
9:         $\mathbf{W}_{:,j+1:i+B} \leftarrow \mathbf{W}_{:,j+1:i+B} - \mathbf{E}_{:,j-i} \cdot \mathbf{H}^{-1}_{j,j+1:i+B}$   ▷ Apply error compensation to future columns
10:    **end for**
11:    $\mathbf{W}_{:,i+B:i+2B} \leftarrow \mathbf{W}_{:,i+B:i+2B} - \mathbf{E} \cdot \mathbf{H}^{-1}_{i:i+B,i+B:i+2B}$       ▷ Final block correction
12: **end for**
13: **return** $\mathbf{Q}$

---

# F    Theoretical Analysis: Why INT4 Quantization Fails

**Setup and notation.**    Let $w \in \mathbb{R}^d$ denote the parameters of a trained model, and $L(w)$ its task loss. We write $H \equiv \nabla^2 L(w)$ for the local curvature (or an empirical-Fisher surrogate). For a uniform signed quantizer with bit width $b$ and scale $s > 0$,

$$B \triangleq 2^{b-1} - 1,$$
$$Q_s^{(b)}(w_j) = \left[ \mathrm{clip}\big(\mathrm{round}(w_j/s), -B, B\big) \right] s, \tag{72}$$

and define the quantized weights $q_j = Q_s^{(b)}(w_j)$ with quantization error

$$\Delta_j \triangleq w_j - q_j. \tag{73}$$

When there is no clipping, $|\Delta_j| \le s/2$; when $|w_j| > Bs$ (saturation), $|\Delta_j| \ge |w_j| - Bs$. We focus on $b = 4$ (*INT4*), where $B = 7$ and the representable range is $[-7s, 7s]$.[3]

**Second-order post-quantization correction.**    Following the quadratic approximation underlying our Shapley-based formulation, we define an effective diagonal curvature term:

$$\tilde{H}_{jj} = \big(\beta w_j + 1 - \beta\big) H_{jj}, \tag{74}$$

and the corresponding elementwise correction

$$E_{:,j} = \frac{w_j - q_j}{\tilde{H}_{jj}}. \tag{75}$$

A one-step second-order refinement updates the weights as

$$w' = w - EH^{-1}. \tag{76}$$

For theoretical clarity, we fix $\beta = 1$ (pure Shapley-weighted curvature), so $\tilde{H}_{jj} = w_j H_{jj}$. Substituting equation 75 into equation 76 and retaining the dominant scaling in $s$ yields

$$w' \approx w - \frac{\Delta}{Hs^2}, \tag{77}$$

where we used that (i) for non-saturated weights $\Delta = \mathcal{O}(s)$, and (ii) $\tilde{H}$ remains comparable in scale to $H$ under a block-diagonal approximation.

---

[3]For the symmetric scheme in equation 72, the maximum index is $B = 2^{b-1} - 1$, yielding a largest representable magnitude of $7s$. Asymmetric variants ($[-8s, 7s]$) differ only by constants.

**Scaling implication.**  From equation 77, the correction magnitude decays quadratically with the scale:

$$\delta w \triangleq w' - w \approx -\frac{\Delta}{Hs^2}.$$

Hence a larger $s$ suppresses curvature-based compensation, while a smaller $s$ amplifies it and risks numerical instability. Empirically, Shapley importance values span a heavy-tailed range of roughly $[10^{-3}, 10^6]$; mapping such a distribution into a few quantization levels magnifies both instability modes discussed below.

**The INT4 dilemma.**  For $b=4$, only $2^4 = 16$ quantization codes are available, corresponding to $2B+1 = 15$ symmetric levels (including zero). To avoid saturating large weights, the scale $s$ must increase until $\max_j |w_j| \leq Bs = 7s$. This makes $\|\Delta\| = \mathcal{O}(s)$ large, while the compensation term in equation 77 shrinks as $1/s^2$:

$$\|\delta w\| \approx \left\| \frac{\Delta}{Hs^2} \right\| \sim \mathcal{O}\left(\frac{1}{Hs}\right),$$

leading to systematic *under-compensation* and residual error. Conversely, choosing a smaller $s$ to reduce rounding error causes widespread clipping ($|w_j| > 7s$), making $\Delta$ a biased truncation term that invalidates the local quadratic approximation. At the same time, the factor $1/(Hs^2)$ can grow excessively, introducing severe numerical instability.

**Interaction with Shapley-driven compression.**  Our compression pipeline leverages Shapley values to prioritize parameter sensitivity. In practice, these scores span orders of magnitude ($[10^{-3}, 10^6]$). INT4's 16 discrete codes cannot represent such variation faithfully: after normalization, many parameters with different importances collapse into identical quantization levels. This dynamic-range collapse forces either (i) a large $s$ to accommodate outliers, or (ii) a small $s$ that incurs severe clipping—thus reproducing the two failure modes described above.

**Comparison with INT8.**  At $b=8$, 256 available codes allow choosing a smaller $s$ that avoids both saturation and instability. The quantization error $\Delta$ remains bounded, the quadratic approximation holds, and the compensation in equation 77 remains accurate and stable. Hence, the very same second-order mechanism that performs well at INT8 becomes ill-conditioned under INT4.

**Conclusion.**  Equations equation 74–equation 77 reveal that INT4 operates in a narrow and fragile regime:

- **Large** $s$: quantization error grows, while curvature-based compensation vanishes ($\propto 1/s^2$) $\Rightarrow$ under-compensation;

- **Small** $s$: clipping dominates and $1/(Hs^2)$ explodes $\Rightarrow$ instability and loss of quadratic validity.

When coupled with heavy-tailed Shapley importance scores, the 16-level INT4 codebook cannot preserve task-relevant sensitivity. Robust INT4 quantization therefore requires additional structure— such as per-channel scaling, outlier preservation, nonlinear remapping of importance, or damped curvature inversion. Absent these, accuracy degradation is unavoidable.

**Discussion: Toward Robust INT4 Quantization.**  The analysis above suggests that the core limitation of INT4 lies in the interaction between coarse quantization steps and the diagonal Hessian approximation. Because $\tilde{H}$ is estimated per-parameter and can vary by several orders of magnitude, its inversion in equation 77 becomes ill-conditioned when combined with a large scale $s$. Future work could therefore aim to *stabilize or regularize the diagonal curvature* before applying second-order corrections. Promising directions include: (i) smoothing $\tilde{H}$ within parameter groups or channels to avoid extreme ratios; (ii) damping or clipping the inverse Hessian term $1/\tilde{H}_{jj}$ to limit numerical explosion; (iii) learning a nonlinear remapping of $\tilde{H}$ that preserves curvature ordering while compressing its dynamic range; and (iv) integrating adaptive scaling tied to local Hessian statistics, so that quantization and curvature correction co-adapt. Such curvature-aware stabilization may enable INT4 quantization to retain second-order fidelity without resorting to mixed-precision designs.

---

**Algorithm 2** Parameter-wise Shapley Value Estimation with Gradient Similarity

---

**Require:** Model parameters $\Theta^\tau = \{\theta_1^\tau, ..., \theta_M^\tau\}$, loss function $\mathcal{L}$, mini-batch size $B$, smoothing coefficient $\alpha$, total steps $T$

**Ensure:** EWMA-based Shapley value estimates $\{\widehat{\phi}_i^T\}_{i=1}^M$

1: Initialize $\widehat{\phi}_i^0 \leftarrow 0$, for all $i = 1,...,M$
2: **for** $\tau = 1$ to $T$ **do**
3:      Sample mini-batch $\mathcal{B}^\tau = \{(x_j, y_j)\}_{j=1}^B$
4:      Compute gradient: $\mathbf{g}^\tau \leftarrow \nabla_\Theta \mathcal{L}_B(\Theta^\tau)$
5:      Compute curvature approximation: $\mathbf{H}^\tau \leftarrow \mathbf{g}^\tau \times \mathbf{g}^\tau$
6:      $\phi^{(1)} \leftarrow -g^\tau \cdot \theta^\tau$
7:      $\phi^{(2)} \leftarrow -\frac{1}{2}\theta^\tau \cdot (H^\tau \times \theta^\tau)$
8:      $\phi^\tau \leftarrow \phi^{(1)} + \phi^{(2)}$
9:      $\widehat{\phi}^\tau \leftarrow (1-\alpha)\widehat{\phi}^{\tau-1} + \alpha \cdot \phi^\tau$
10: **end for**
11: **return** $\{\widehat{\phi}_i^T\}_{i=1}^M$

---

# G    Algorithm and Time Complexity

**Algorithm 2.**    This algorithm computes parameter-wise Shapley values using gradients and curvature approximations.

Line 1 initializes the exponentially weighted moving average (EWMA) estimate for each parameter. Lines 2–5 iterate through training steps, where at each iteration, the algorithm samples a mini-batch and computes the parameter gradient $\mathbf{g}^\tau$ and an approximate curvature matrix $\mathbf{H}^\tau$ (using Fisher approximation). Lines 6–9 compute the per-parameter Shapley contributions: line 6 calculates the individual importance via first-order terms; lines 7 account for self-curvature and parameter interaction terms using second-order curvature information. The total Shapley score is then aggregated in line 8, and line 9 updates the running average using the EWMA rule. Finally, line 11 returns the smoothed Shapley scores after $T$ steps.

**Time and Space Complexity.**    The overall computational cost per step is dominated by the gradient calculation and the curvature approximation. Assuming the size of parameters in the model is $M$ and the mini-batch size is $B$, the gradient step costs $\mathcal{O}(B \cdot M)$, while the full curvature matrix requires up to $\mathcal{O}(M^2)$ operations. If blockwise Fisher approximation is used with block size $d$, the cost can be reduced to $\mathcal{O}(M \cdot d)$. The total space complexity is also $\mathcal{O}(M^2)$ in the worst case, or $\mathcal{O}(M \cdot d)$ when using blockwise regularization. This makes the algorithm scalable in practice for modern networks with structured parameter grouping.

# H    More Experiment Details

## H.1    Datasets Statics

We employ four benchmark datasets across natural language understanding and computer vision domains to evaluate the generality and robustness of our method. The detailed statistics and task settings are summarized below.

(1) GSM8K (Grade School Math 8K) [62] is a dataset of 8.5K high quality linguistically diverse grade school math word problems. This dataset is usually used to evaluate the perofrmance of language models on basic mathematical problems that require multi-step reasoning.

(2) MMLU (Massive Multitask Language Understanding) [63] is a benchmark designed to evaluate models' multitask language understanding across 57 tasks, covering subjects such as humanities, STEM, social sciences, and more. Each task consists of multiple-choice questions, typically with four options. The dataset is split into a development and test set, with no training set provided. This benchmark is primarily used for evaluating zero-shot or few-shot performance in language models, focusing on factual reasoning and knowledge retention.

(3) CIFAR-100 [64] is a widely used image classification dataset consisting of 60,000 32×32 color images in 100 classes, with 600 images per class. Among the 600 images in each class, there are 500 training images and 100 testing images. It is a standard benchmark for evaluating image classification performance, particularly in the context of resource-efficient training, model compression, and continual learning.

(4) ImageNet [65] is a large-scale image classification dataset with over 1.2 million training images and 50,000 validation images across 1,000 object categories. It is one of the most prominent benchmarks in computer vision, used to assess the representation power, generalization, and transferability of visual models. The dataset serves as a foundation for pretraining and evaluating deep convolutional and transformer-based vision models. We use the release version of ImageNet in 2012 as the dataset in our experiments.

## H.2 Baselines Details

In this section, we introduce the baselines we used for quantification of important parameters in NLP and CV tasks, and the baselines that are widely used in LLM quantization.

For NLP and CV tasks, we systematically compare our method with the following baseline strategies to quantify the contribution of each neuron in the model.

- **Random Masking** This baseline assigns uniform random importance scores to neurons without considering their structural or functional relevance. Specifically, each neuron $n_i$ in the network is assigned a score $s_i \sim \mathcal{U}(0,1)$ from a uniform distribution. This strategy establishes a non-informative reference to other quantification methods.
- **Gradient Magnitude** [43, 5] This baseline prioritizes neurons based on the magnitude of their gradient values $|\mathbf{g}_i^\tau|$ computed during backpropagation. Specifically, given a parameter $\theta_i^\tau$ in the training step $\tau$, its importance score is defined as the absolute gradient $s_i = \left| \frac{\partial \mathcal{L}}{\partial \theta_i^\tau} \right|$, where $\mathcal{L}$ denotes the loss function. This strategy assumes that parameters with larger gradients contribute more significantly to reducing the loss and thus deserve higher retention priority during pruning or masking.
- **Gradient Trace** [40, 35, 49, 43] This method computes importance scores by the product of parameter values and their corresponding gradients, i.e., $s_i = |\theta_i^\tau \cdot \mathbf{g}_i^\tau|$. The intuition is to capture both the sensitivity ($\mathbf{g}_i^\tau$) and the scale ($\theta_i^\tau$) of each parameter, effectively identifying neurons whose activation patterns are both volatile and impactful. By incorporating parameter magnitudes, this approach mitigates the bias towards large gradients in low-magnitude parameters, providing a more balanced measure of importance.

For quantization tasks, we compare our method with GPTQ [4] and OBD [38]. The details of these two methods are shown below.

- **GPTQ** [4] This method represents a state-of-the-art post-training quantization (PTQ) technique designed for large language models. GPTQ employs a greedy Hessian-based algorithm to quantize each parameter group $\theta_i$ independently. Specifically, it minimizes the quantization error $\|\theta_i - \hat{\theta}_i\|_F^2$ under the constraint that $\hat{\theta}_i$ lies within the specified quantization level (e.g., 4-bit or 8-bit). The optimization is formulated as:

$$\hat{\theta}_i^* = \arg\min_{\hat{\theta}_i \in \mathcal{Q}} \|\theta_i - \hat{\theta}_i\|_F^2 + \lambda \cdot \mathrm{tr}(\hat{\theta}_i^T \mathbf{H}_i \hat{\theta}_i) \tag{78}$$

where $\mathcal{Q}$ denotes the discrete quantization space, $\mathbf{H}_i$ is the approximate Hessian matrix capturing the sensitivity of the output to each parameter, and $\lambda$ is a regularization parameter. This group-wise quantization approach efficiently balances accuracy and computational overhead.

- **OBD** [38] Optimal Brain Damage (OBD) is a gradient-based pruning and quantization framework that leverages the second-order Taylor expansion of the loss function. Given a parameter $\theta_{ij}$ at the $i$-th layer and $j$-th position, OBD computes the saliency score as:

$$S(\theta_{ij}) = \frac{1}{2} \cdot \theta_{ij}^2 \cdot \frac{\partial^2 \mathcal{L}}{\partial \theta_{ij}^2} \tag{79}$$

During quantization, parameters with lower saliency scores are prioritized for quantization or removal, as they contribute less to the overall loss. OBD typically uses a Hessian diagonal

approximation to efficiently estimate the second-order derivatives, enabling scalable application to large models. Unlike GPTQ, OBD integrates pruning and quantization decisions, making it suitable for joint model compression tasks.

We evaluate different quantization settings commonly used in model compression, including INT4, INT8, and FP8, as detailed below:

- **INT4 / W4A16** quantization uses 4-bit integers to represent model weights. In the W4A16 setting, weights (**W**) are quantized to 4 bits, while activations (**A**) remain in 16-bit floating point (FP16). This configuration achieves significant model size and memory bandwidth reduction while maintaining inference accuracy due to high-precision activations.

- **INT8 / W8A8** quantization is a widely adopted precision format where both weights and activations are quantized to 8-bit integers. The W8A8 configuration balances compression and performance, offering compatibility with many hardware accelerators and maintaining near-floating-point accuracy in many practical applications.

- **FP8** is a low-precision floating-point format (e.g., E4M3 or E5M2) gaining traction for training and inference. While not strictly integer quantization, FP8 enables dynamic range advantages over INT formats. It can be used to represent either weights or activations or both, depending on the hardware support. Some hybrid schemes combine INT8 for weights with FP8 for activations or vice versa, leveraging the benefits of both representations.

### H.3  Prompts used in MODEL SHAPLEY

In this section, we provide a detailed display of all the prompts used.

- **Prompt for MMLU Dataset** The following prompt is leveraged to instruct LLM to answer MMLU dataset questions.

> **Prompt 1**
>
> The following is a multiple choice question. Think step by step and then finish your answer with "the answer is (X)" where X is the correct letter choice.
> **[Question]**    {*Question q*}
> **A**. {*Content_of_A*}
> **B**. {*Content_of_B*}
> **C**. {*Content_of_C*}
> **D**. {*Content_of_D*}

- **Prompt for GSM8K Dataset** The following chain-of-thought prompt is used to generate step-by-step answers [66] to instruct LLM to answer GSM8K dataset questions.

> **Prompt 2**
>
> {*Question q*} Let's think step by step and output the final answer after "####".

- **Prompt for Private Company Rules Dataset** The following prompt is used to generate answers by selecting the most relevant knowledge base.

> **Prompt 3**
>
> {*Question q*} You are a domain-specific large language model integrated with multiple knowledge bases. The list of knowledge bases is as follows: {database_list}
>
> # Please analyze the question "{query}" and determine which knowledge base it most likely belongs to. If none of them are appropriate, output "others".

By using these prompts above to guide the LLM to output the final answer in a specified format, we can then extract the model's output through regular expression matching and conduct evaluations.

## H.4 Implementation Details

We employ three different parameter-level language model architectures as our backbones, including *Qwen2.5-3B-Instruct*, *Qwen2.5-7B-Instruct* [57], and *LLaMA-3.2-3B-Instruct* [32]. We use PyTorch 2.6.0 library to implement all the algorithms based on the open-source HuggingFace transformers [67] codebase. All NLP tasks and CV tasks are tested on an Ubuntu server equipped with 8 NVIDIA A100 GPUs with 80GB memory, and quantization tasks are tested on an Ubuntu server equipped with 8 NVIDIA 4090 GPUs with 48GB memory. During the deactivation process, we only care about MLP layers in the transformer models. And we deal with the weight and bias of a neuron at the same time.

For the training experiments, The activative ratio is set to 0.1 for all tasks. We conduct NLP tasks with a learning rate of 3e-5, a max gradient norm of 1.0, a warmup ratio of 1e-3, a batch size of 16 for *GSM8K* dataset and 8 for *MMLU* dataset, a gradient accumulation steps of 4, a cutoff length of 1024 for *GSM8K* dataset and 2048 for *MMLU* dataset, a max respoonse length of 1024, and 1 epochs. And CV tasks are conducted with a learning rate of 1e-5, a weight decay of 1e-4, a batch size of 512, max epochs of 50 and use early stop strategy with patience of 5 epochs.

For the inference experiments, we set activative ratio to 0.95 for all NLP tasks, and we set activative ratio to 0.7 for CV tasks. The batch size setting is the same as the train experiments.

For the quantization experiments, we set the calibration sample number to 512, and max input sequence length to 2048. All quantization experiments are conducted on *GSM8K* dataset.

## H.5 Deactivate Neurons' Case Study

In this section, we present a case study to visually demonstrate the effectiveness of **MODEL SHAPLEY**. As shown in Table 6, Table 7, Table 4, Table 5, we analyze the Qwen 2.5 7B-Instruct model on the MMLU and GSM8K datasets, evaluating the impact on model outputs after deactivating the bottom-K neurons (5%, 10%, and 30%) identified based on their **MODEL SHAPLEY** values. These key neurons correspond to functional units associated with knowledge retrieval and mathematical reasoning tasks.

Under normal conditions, the model demonstrates strong performance in both factual response generation and mathematical reasoning. When 5% of the key neurons are deactivated, the model still produces reasonable outputs. However, upon deactivating 10% of the key neurons, we observe a noticeable decline in output quality, with significant drops in both fluency and accuracy. At 30% deactivation, the model begins to generate incoherent or irrelevant responses, effectively losing basic conversational capabilities.

These results provide compelling evidence for the effectiveness of **MODEL SHAPLEY** in identifying critical neurons responsible for specialized cognitive functions.

## H.6 Deactivation Ration Study

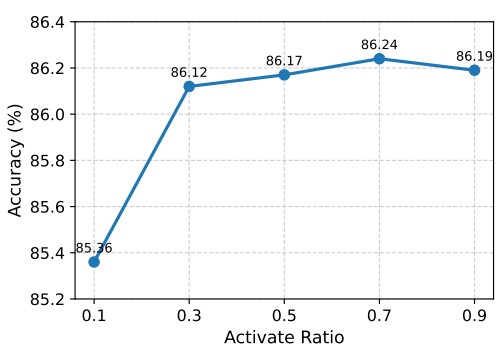

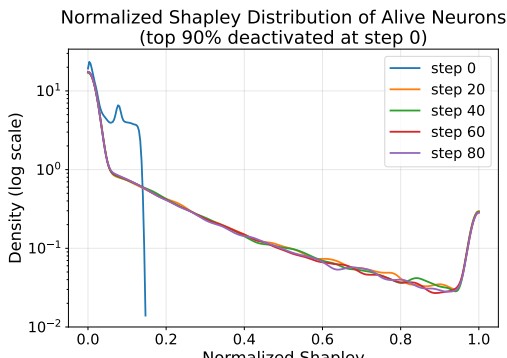

(a) Accuracy on CIFAR-100 of VIT-Base-16 Backbone at different activation ratios during finetune.

(b) Compensation Study of Qwen 2.5-3B Instruct on GSM8K Dataset.

We conducted experiments on the ViT-Base-16 model using the CIFAR-100 dataset. As shown in Figure 3a, the results reveal that a significant portion of the neurons are redundant. Remarkably,

Table 4: Case study on MMLU Dataset comparing the normal output and deactivated BottomK output.

| Question | **The numen of Augustus referred to which of the following characteristics? The choices are: A. Divine power, B. Sexual virility, C. Military acumen, D. Philosophical intellect** |
|---|---|
| **Ground Truth** | **(A) Divine power** |
| **Normal Output** | To determine the correct answer, let's consider what numen meant in the context of Roman religion and the time of Augustus. Numen was a concept in Roman religion that referred to the distinctive power or presence of a deity. It was not a personification but rather a kind of spiritual presence that could be associated with an object or place. In the context of Augustus, who was elevated to near-divine status by the Roman people and Senate, his numen would refer to his divine power or the divine presence attributed to him.- Option A: Divine power aligns closely with the concept of numen in this context.- Option B: Sexual virility does not fit the general notion of numen. - Option C: Military acumen, while a characteristic that made Augustus highly successful, is not specifically related to the concept of numen.- Option D: Philosophical intellect, again, does not relate specifically to numen. Thus, the numen of Augustus referred to his divine power.The answer is (A). **[CORRECT]** |
| **Deactivated Output (5%)** | To determine the correct answer, let's consider what ïumenïn the context of a historical figure refers to.1. **Divine power (A)**: In Roman mythology and history, the numen often referred to divine forces or their manifestations. Augustus was given a numen to indicate a divine presence or power that could protect and influence the empire.2. **Sexual virility (B)**: While sexual virility might have been a characteristic of Roman leaders for historical or symbolic purposes, it doesn't typically align with the term ïumenïn historical contexts.3. **Military acumen (C)**: While military acumen might have been a significant trait for leaders, the term ïumenïs more aligned with powers attributed to deities or divine influence rather than specific human traits like military skills.4. **Philosophical intellect (D)**: Similarly, philosophical intellect would relate more to mental prowess rather than the divine or protective aspect typically associated with a numen.The term ïumenïn the context of Roman emperors and Augustus specifically relates to divine power or a divine presence, which is a characteristic that could protect and influence the empire on a supernatural level.Therefore, the answer is (A). **[CORRECT]** |
| **Deactivated Output (10%)** | To determine the correct answer, let's consider what ïumenïn the context of Augustus refers to. The term ïumenïs associated with the Roman gods and can be linked to the concept of divine attributes or powers within the Roman context.In ancient Rome, a ïumenöuld be the power or the divine aspect of a god. An individual or deity could be associated with numen, which would manifest their divine attributes or powers.Let's evaluate each option:A. Divine power - This option aligns well with the concept of a numen. A numen would indeed refer to a divine power or attribute that is associated with a divine entity.B. Sexual virility - This is not relevant to the concept of numen in ancient Rome. It might perturb the context to something more related to ancient Greek, where attributes like virility were discussed within other deities, such as Priapus.C. Military acumen - While military acumen can be a trait of a leader, it is not directly related to the concept of numen, which pertains more to divine qualities.D. Philosophical intellect - Although a philosophical intellect could be a characteristic, it is not directly related to the term ïumen,̈which is more aligned with divine attributes.Given the context and the specific meaning of ïumenïn relation to divine attributes, the answer is:A. Divine power. The answer is (X) where X is the correct letter choice A. **[CORRECT]** |
| **Deactivated Output (30%)** | To answer this question step, you must follow the process of a number in the reference of a m The correct answer is C, M̈ilitary acumen.̈The answer is (C).Step reason:The current is in a 1 5 1 5 1 5 the 8 An acuen of acuen of U A F 1 5 1 5 1 A the of 5 1 1 Relevant M 1 5-With with 5 5 5 5 5 5 5 5 the 5 rule 1 ab 5 5 5 5 u 5-shift 1 an the (*(1 5 the 5 The 5 The 5 5 The 5 the 5) the 5 5 a 5 This A comes the 8 5 the 5 Prevention 52 2 1 1 1 1 1 1 5 3 1 1 D 1 1 1 1 1 1 1 1 1 1 1 1 1 1 the 2 5 1 a 1 5 1 1 M u 1 1 1"] **[WRONG]** |

Table 5: Case study on MMLU Dataset comparing the normal output and deactivated BottomK output.

| | |
|---|---|
| **Question** | **Every country with a seacoast has an exclusive economic zone (EEZ) under the UN Convention on the Law of the Sea. Which one of the following statements regarding the Law of the Sea and the EEZ is NOT correct?" Choices": A. "The EEZ extends up to 200 nautical miles (370 km) from the coastline.", B. "Countries have the right to exploit natural resources in their EEZ.", C. "Countries can exploit resources on the continental shelf extending up to 350 nautical miles (560 km) beyond their EEZ.", D. "A country does not have the right to exploit living resources in its EEZ".** |
| **Ground Truth** | **(D) A country does not have the right to exploit living resources in its EEZ** |
| **Normal Output** | Let's analyze each statement regarding the Law of the Sea and the EEZ:A. The EEZ extends up to 200 nautical miles (370 km) from the coastline. According to the UN Convention on the Law of the Sea, this statement is correct.B. Countries have the right to exploit natural resources in their EEZ. - This statement is also correct as it aligns with the principle of the EEZ. C. Countries can exploit resources on the continental shelf extending up to 350 nautical miles (560 km) beyond their EEZ.- This statement is correct, as states have the right to explore and exploit all natural resources of the seabed and subsoil beyond and adjacent to their territorial sea up to 350 nm from the baselines. D. A country does not have the right to exploit living resources in its EEZ. - This statement is incorrect. Countries have the right to explore and exploit living resources within their EEZ, contrary to what D claims. After reviewing all the statements, we find that only one is incorrect. Therefore, the answer is (D). **[CORRECT]** |
| **Deactivated Output (5%)** | To determine which statement is NOT correct, let's analyze each option step by step: A. The EEZ extends up to 200 nautical miles (370 km) from the coastline. This statement is correct according to the UN Convention on the Law of the Sea, which defines the EEZ extending up to 200 nautical miles. B. Countries have the right to exploit natural resources in their EEZ. This statement is also correct. Article 77 of the convention states that states have the authority to exploit resources in their EEZ for reproduction purposes. C. Countries can exploit resources on the continental shelf extending up to 350 nautical miles (560 km) beyond their EEZ. This statement is accurate under the convention as it pertains to the continental shelf, which can extend up to 350 nautical miles beyond the EEZ, as mentioned in Article 76. D. A country does not have the right to exploit living resources in its EEZ. This statement is incorrect. Contrary to what D suggests, Article 105 of the convention actually allows states to exploit living resources in their EEZ, subject to certain conditions. The answer is (D). **[CORRECT]** |
| **Deactivated Output (10%)** | Let's analyze each statement regarding the Law of the Sea and the Exclusive Economic Zone (EEZ):A. The EEZ extends up to 200 nautical miles (370 km) from the coastline. This statement is correct according to the Convention on the Law of the Sea. The EEZ is the area extending from the coastline to 200 nautical miles. B. Countries have the right to exploit natural resources in their EEZ. This statement is also correct. Countries can exploit natural resources in the EEZ, including renewable resources, provided they do not leave a negative impact on the marine environment. C. Countries can exploit resources on the continental shelf extending up to 350 nautical miles (560 km) beyond their EEZ. This statement is not correct according to the Convention. The continental shelf extends up to 100 nautical miles (350 km) from the coastline, while the EEZ extends to 200 nautical miles from the coastline. The continental shelf is not part of the EEZ and extends only up to 100 nautical miles (350 km). D. A country does not have the right to exploit living resources in its EEZ. This statement is correct according to the Convention. While countries can exploit natural resources in the EEZ, living resources are not included in the natural resources under the Convention. The statement that is not correct is C, as it incorrectly states the continental shelf extends beyond the EEZ, which is not true. The answer is (C)." **[WRONG]** |
| **Deactivated Output (30%)** | The correct answer is the answer (A) and the to the in you and the a in you in in Mor the years D in in in in in in in in in in in in in in in the (The answer is (D)) is in in Condition (2 in n, 8 the in in in in the ( in in 2 2 in the 2 The 2 in in 1 1 A in in 3 1 1 1 1 1 in in 1 1 1 1 1 1 1 1 1 the 1 1 1 1 1 1 1 1 1 the 1 1 1 1 1 1 1 1 1 1 1 3 3 3 **[WRONG]** |

Table 6: Case study on GSM8K Dataset comparing the normal output and deactivated BottomK output.

| Question | Josh decides to try flipping a house. He buys a house for $80,000 and then puts in $50,000 in repairs. This increased the value of the house by 150%. How much profit did he make? |
|---|---|
| Ground Truth | 70000 |
| Normal Output | Let's break down the problem step by step:
1. **Initial Cost**: Josh buys a house for $80,000.
2. **Repairs Cost**: He spends $50,000 on repairs.
So, the total cost including repairs is:
$80,000 + 50,000 = 130,000$ dollars
3. **Value Increase**: The repairs increase the value of the house by 150%.
To find the increase in value, we calculate 150% of the purchase price:
$150\% \times 80,000 = 1.5 \times 80,000 = 120,000$ dollars
So, the new value of the house is:
$80,000 + 120,000 = 200,000$ dollars
4. **Profit Calculation**: The profit is the selling price minus the total cost.
$Profit = 200,000 - 130,000 = 70,000$ dollars
$\boxed{70,000}$ **[CORRECT]** |
| Deactivated Output (5%) | Let's break down the problem step by step:
1. **Initial Purchase**: Josh buys a house for $80,000.
2. **Repairs**: Josh spends an additional $50,000 on repairs.
3. **Total Investment**: The total amount Josh has invested (purchase price + repairs) is:
$80,000 + 50,000 = 130,000$
4. **Increase in Value**: The repairs increased the value of the house by 150%. To find the new value of the house, we need to calculate 150% of the purchase price and add it to the purchase price:
Increase in value $= 80,000 \times 1.50 = 120,000$
So, the new value of the house is:
$80,000 + 120,000 = 200,000$
5. **Profit Calculation**: Profit is calculated by subtracting the total investment from the new value of the house:
$Profit = 200,000 - 130,000 = 70,000$
Josh made a profit of $70,000. **[CORRECT]** |
| Deactivated Output (10%) | Step 1: Calculate the total cost of the house, including repairs The house was bought for $80,000 and then $50,000 was spent on repairs, so the total cost is: $80,000 + 50,000 = 130,000$ Step 2: Determine the increased value of the house The value of the house increased by 150% after the repairs, so the new value of the house is: $1.5 \times 80,000 = 130,000$ In this case, 1.5 times the original value of the house is equal to 130,000, indicating that the original value of the house was $80,000 and the repairs added $100,000 to the value (150% - 1 = 1.5, means 1 times the original plus 1 times the, so 130,000 - 130,000 = 100,000). Step 3: Calculate the profit The profit is the difference between the increased value of the house and the total cost of the house, so The profit is the increased value (130,000) minus the cost (130,000), so 130,000 (150% - 130,000), so 130,000 (150% - 130,000) = 130,000 (150% - 130,000) = 130 **[WRONG]** |
| Deactivated Output (30%) | Let's think through the steps to calculate the profit of the house, in the context of a flip and the increase of the value by 150%. Step by step, we'll output the final answer: Step 1: Initial Value of the House Let's start with the initial value of the house - Initial value: $80,000 To the 150% increase, we need to a calculate. The 150% increase is a step, and it is the value that we need to calculate. Let's assume the initial value is a 150% increase, for the purpose of the 150% increase, to the 150% increase itself the 150% increase is a 150% increase in the current value of the technology, not the 150% increase in the initial value itself. This, we will calculate the increase of the 150% value as a 150% increase in the next value, and the 150% increase in the 1 55 5 5 the 2 1 5 1 1 1 1 1 1 1 1 1 1 1 1 the, 1 with the, 1 1 Step 2: Calcul with the 150% Increase In the 1 5 5 5, addon, , Let, the, the, 1 5, Let, this, th for let ihm', the, the, the, the, ' 5' atte', API', **[WRONG]** |

Table 7: Case study on GSM8K Dataset comparing the normal output and deactivated BottomK output.

| | |
|---|---|
| **Question** | **Janet's ducks lay 16 eggs per day. She eats three for breakfast every morning and bakes muffins for her friends every day with four. She sells the remainder at the farmers' market daily for $2 per fresh duck egg. How much in dollars does she make every day at the farmers' market?** |
| **Ground Truth** | **18** |
| **Normal Output** | Let's break down the problem step by step:
1. **Calculate the total number of eggs Janet gets each day:**
- Janet's ducks lay 16 eggs per day.
2. **Calculate the number of eggs Janet uses for breakfast:**
- Janet eats 3 eggs for breakfast every morning.
3. **Calculate the number of eggs Janet uses for baking muffins:**
- Janet uses 4 eggs to bake muffins for her friends every day.
4. **Calculate the total number of eggs Janet uses:**
- Total eggs used for breakfast and baking = 3 (breakfast) + 4 (muffins) = 7 eggs per day.
5. **Calculate the number of eggs Janet can sell:**
- Remaining eggs after usage = Total eggs per day - Eggs used
- Remaining eggs = 16 - 7 = 9 eggs per day.
6. **Calculate the revenue from selling the eggs:**
- Janet sells the remaining 9 eggs at $2 per egg.
- Revenue = 9 eggs * $2/egg = $18 per day.
$\boxed{18}$ **[CORRECT]** |
| **Deactivated Output (5%)** | Let's break down the problem step by step:
1. **Determine the total number of eggs laid per day:**
- Janet's ducks lay 16 eggs per day.
2. **Calculate the number of eggs used for breakfast:**
- Janet eats 3 eggs for breakfast every morning.
- So, the number of eggs used for breakfast is 3 eggs.
3. **Calculate the number of eggs used for baking muffins:**
- Janet bakes muffins using 4 eggs each day.
- So, the number of eggs used for baking is 4 eggs.
4. **Calculate the total number of eggs used (for breakfast and baking):**
- The total number of eggs used is $3+4=7$ eggs.
5. **Calculate the number of eggs remaining for selling:**
- The number of eggs remaining after using some for breakfast and baking is $16-7=9$ eggs.
6. **Calculate the daily earnings from selling the remaining eggs:**
- Janet sells the remaining eggs at $2 per egg.
- So, the daily earnings are $9 \times 2 = 18$ dollars.
Therefore, Janet makes 18 dollars every day at the farmers' market. **[CORRECT]** |
| **Deactivated Output (10%)** | To solve this problem, let's first calculate the number of eggs Janet has after various activities and then determine the earnings from selling the remainder.
1. **Total eggs laid by Janet's ducks in a day (16 eggs)**
2. Eggs eaten for breakfast (16 eggs - 3 eggs for the morning breakfast = 13 eggs)
3. Eggs used for baking muffins (16 eggs - 4 eggs for baking muffins = 12 eggs)
4. Eggs sold at the farmers' market (16 eggs - 3 eggs for morning breakfast + 4 eggs for baking - 2 eggs = 8 eggs)
3. **Earnings per egg at the farmers' market (2 dollars per egg)**
4. **Earnings calculation (8 eggs multiplied by 2 dollars per egg = 16 dollars)**
So, Janet makes 16 dollars every day at the farmers' market by selling the remainder of her eggs. $\boxed{16}$ **[WRONG]** |
| **Deactivated Output (30%)** | Let's break the problem step by step and calculate the number of the daily you would make for the visit to the farmers's market.
Step by step:
1. Calculate the number of the daily Mor's (d and " in) in the first, to the work of the Mor ( ( in. to the 8 and " 0 0))):
The Mor ( ( ( ( ( ) 16 4 ( 0 ( ) ) ( ) ). 2 ( ( in). to the ...the the ( ( the ( ed ( ( the ( M !! ! ! ( ) !! !) ! 2 ( !) ( it (**[WRONG]** |

retaining only the top 70% of neurons ranked by their Shapley values achieves an accuracy of 86.42%, surpassing the 85.19% accuracy obtained when 90% of the neurons are retained.

This finding suggests that not all neurons contribute equally to the model's predictive capability, and that a carefully selected subset of high-importance neurons can maintain or even enhance performance. It also highlights the potential for neuron pruning or activation sparsity as an effective strategy for improving efficiency without sacrificing accuracy.

### H.7 Analysis of Compensation Effects in Parameter Units

During our experiments, we observed the **neural compensation effect**. To further investigate this phenomenon, we design an additional set of experiments using the Qwen 2.5 3B-Instruct model on the GSM8K dataset. Specifically, at various training steps (step = 0, 20, 40, 60, and 80), we deactivated the top 90% of neurons ranked by importance and analyzed the distribution of parameter importance.

Our results in Figure 3b show that as training progresses, the distribution of Shapley values evolves. When high-Shapley neurons are deactivated, other neurons progressively assume their functional roles. This redistribution leads to increased Shapley values among previously less important neurons, illustrating a clear compensation effect within the network. These findings suggest that the model dynamically redistributes functional importance across neurons during training to adapt to perturbations, highlighting the inherent resilience and adaptability of neural representations.

### H.8 Model Compression Sensitivity Experiments

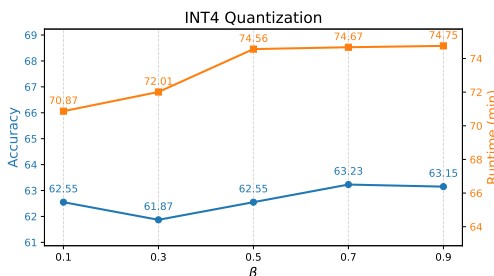 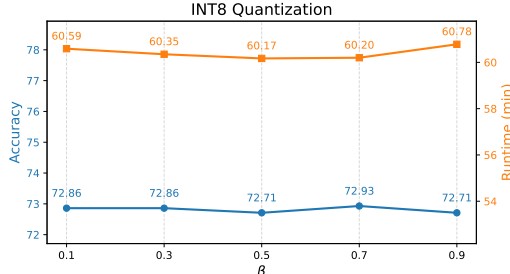

(a) INT 4 Quantization with different $\beta$ on GSM8K.  (b) INT 8 Quantization with different $\beta$ on GSM8K.

We further compared the performance of Qwen 2.5 7B-Instruct and Qwen 2.5 14B-Instruct under different compression settings for INT 4 in Figure 4a and INT 8 in Figure 4b. By varying the parameter $\beta$ in the range $\{0.1, 0.3, 0.5, 0.7, 0.9\}$, we evaluated both time efficiency and model accuracy.

The experimental results demonstrate the effectiveness of **MODEL SHAPLEY**, particularly in scenarios where cooperative game-theoretic contributions are explicitly weighted. As $\alpha$ increases, the trade-off between compression and performance becomes more pronounced, yet **MODEL SHAPLEY** consistently maintains superior accuracy with reduced computational cost. These findings validate the importance of incorporating cooperative importance measures in guiding neuron selection for model compression.

### H.9 Attention Head and Layer-wise MODEL SHAPLEY Study

In this experiment, we investigate the effectiveness of **MODEL SHAPLEY** at the attention head and layer level. Specifically, we evaluate its ability to identify important parameter units beyond MLP layers. We conduct knowledge base routing experiments on a private company policy dataset, following the setup in [68, 69, 70]. The dataset consists of 2,088 training samples and 344 test samples, covering four subcategories of company regulations and an "others" category (i.e., questions that should not trigger knowledge base queries).

The routing task aims to determine whether the model should query a knowledge base and, if so, which one. We compare the performance of **MODEL SHAPLEY** and gradient trace in identifying important attention heads and layers. Using the Qwen 2.5 7B-Instruct model, we compute the Top-K importance scores using both methods and select the most relevant units accordingly.

For probing, we use the training set to identify the most important attention head and layer, and then use the output signals from that selected layer as input to a downstream probing model—a two-layer MLP classifier trained to perform 5-way classification.

Table 8: Top-5 Important Attention Heads and Layers Identified

| Rank | Gradient Trace (Layer : Score) | MODEL SHAPLEY (Layer : Score) |
|------|-------------------------------|-------------------------------|
| **1st** | **model.layers.22.self_attn.v_proj :** $1.90 \times 10^{-5}$ | **model.layers.26.mlp.down_proj** : 0.375 |
| 2nd | model.layers.3.self_attn.v_proj : $1.66 \times 10^{-5}$ | model.layers.25.self_attn.v_proj : 0.354 |
| 3rd | model.layers.25.self_attn.v_proj : $1.52 \times 10^{-5}$ | model.layers.23.self_attn.v_proj : 0.352 |
| 4th | model.layers.26.self_attn.v_proj : $1.44 \times 10^{-5}$ | model.layers.22.self_attn.v_proj : 0.289 |
| 5th | model.layers.23.self_attn.v_proj : $1.40 \times 10^{-5}$ | model.layers.21.self_attn.v_proj : 0.260 |

Based on these rankings, we use model.layers.22.self_attn.v_proj for the gradient trace experiment and model.layers.26.mlp.down_proj for the **MODEL SHAPLEY** experiment. The classification accuracy on the routing task is summarized in Table 9.

Table 9: Comparison of Gradient Trace & **MODEL SHAPLEY** on Attention Head and Layer-wise Probing

| Method | Train Set Accuracy | Test Set Accuracy |
|--------|-------------------|-------------------|
| Gradient Trace (model.layers.22.self_attn.v_proj) | 0.89 | 0.81 |
| **MODEL SHAPLEY** (model.layers.26.mlp.down_proj) | **0.92** | **0.83** |

The experimental results demonstrate that **MODEL SHAPLEY** is effective not only at the neuron level, but also at the finer-grained attention head and layer level.

Table 10: Accuracy (%) and training cost (same units across methods; lower is better) on GSM8K.

| Method | Qwen2.5-3B-Instruct | | Qwen2.5-7B-Instruct | |
|--------|--------|--------|--------|--------|
| | Accuracy | Cost time | Accuracy | Cost time |
| Shapley | 49.51 | 215 | 62.47 | 265 |
| LoRA | 48.45 | 200 | 64.14 | 170 |
| Shapley+LoRA | 52.16 | 242 | 63.53 | 273 |

## H.10 Runtime-normalized comparison

It is crucial to first clarify **where the computational cost occurs**. The runtime difference between our method and the baselines lies in the **one-time, offline parameter importance quantification** step. The subsequent quantization process, guided by these importance scores, takes a similar amount of time for all methods.

Therefore, a constant runtime comparison for the importance quantification step is not straightforward. Each method is a distinct algorithm designed to run to completion. Artificially truncating our method to match a faster baseline's runtime would mean using an incomplete Hessian approximation, compromising the integrity of our approach and leading to an uninformative comparison.

Instead, we believe the right way to evaluate this trade-off is to compare the final accuracy against the actual, practical runtime. We admit our method has a higher cost for importance quantification, as it incorporates second-order cooperative interactions, not just first-order individual importance. Our experiments show this additional investment is both **marginal and highly worthwhile**.

For Qwen2.5-7B INT8 quantization: ❶ GPTQ: 53.86 min total (importance: 5 min), ❷ OBD: 57.88 min total (importance: 8 min), ❸ **MODEL SHAPLEY** : 60.20 min total (importance: 10 min).

The 2-5 minute difference in importance estimation yields 1.5-2% accuracy gains, which persists throughout the model's deployment lifetime. We believe the one-time cost of 5 extra minutes during quantization is negligible compared to the perpetual benefits of higher accuracy and/or smaller model size during deployment, where models serve millions of requests.

## H.11 Fine-tuning Efficiency and Variance

We extend our significance analysis with two compact replications across NLP and vision. In each setting we report mean accuracy and standard error, and we assess statistical significance with independent-samples $t$-tests at the 5% level. We also examine stability across random seeds under realistic compute budgets.

On GSM8K with `Qwen2.5-3B-Instruct`, each configuration is repeated with three random seeds. This replication compares **MODEL SHAPLEY** with the strongest baseline, *Gradient Trace*. Table 11 presents the three-seed statistics. **MODEL SHAPLEY** attains a higher mean and a smaller standard error than *Gradient Trace*, and $t$-tests at the 5% level indicate statistically significant improvements, supporting reliable gains under the three-seed regime.

On CIFAR-100 with `VIT-Base`, we run eight seeds and include four methods: Random, Gradient, Gradient Trace, and **MODEL SHAPLEY**. Table 12 reports the aggregate mean and standard error for these methods. **MODEL SHAPLEY** achieves the highest mean accuracy, while standard errors differ across baselines. Using the full eight-seed replication for inference, pairwise $t$-tests at the 5% level confirm significant improvements of **MODEL SHAPLEY** over each baseline.

Table 11: Accuracy statistics (%) across three runs on GSM8K with `Qwen2.5-3B-Instruct`.

| Method | Mean | Standard Error |
|---|---|---|
| Gradient Trace | 48.49 | 0.72 |
| **MODEL SHAPLEY** | **49.51** | **0.27** |

Table 12: Accuracy statistics (%) across eight runs on CIFAR-100 with `VIT-Base`.

| Method | Mean | Standard Error |
|---|---|---|
| Random | 84.59 | 0.13 |
| Gradient | 85.16 | 0.16 |
| Gradient Trace | 85.12 | **0.10** |
| **MODEL SHAPLEY** | **85.51** | 0.19 |

**Environment note.** The mean values reported here may differ slightly from those in the main text because these additional runs were executed on a different hardware/software stack (NVIDIA H800 GPUs with a newer CUDA environment) than the original experiments (NVIDIA A100 GPUs with a previous CUDA version), yielding a general upward shift in accuracy for all methods.

**Computational efficiency.** **MODEL SHAPLEY** estimates parameter importance with a single forward and backward pass plus a Hessian–vector product (see Section 4.1 and Remark 4.9). For multi-epoch fine-tuning, this one-time cost is amortized by updating only a small subset of parameters. In our experiments, selecting $10\%$ of parameters reduced training time by approximately $85\%$ relative to full-model fine-tuning and reduced gradient storage by $90\%$, enabling larger batch sizes. The one-time importance estimation for a 7B model takes roughly 30 minutes and can be reused across fine-tuning tasks.

## H.12 Hardware Scalability and General Applicability

The scalability of **MODEL SHAPLEY** is primarily governed by the memory and compute requirements of the blockwise Hessian approximation rather than any hardware-specific constraint. The method is designed to be hardware-agnostic and can operate efficiently across diverse GPU architectures by adjusting the block granularity used in Shapley value estimation.

The total memory footprint scales quadratically with the number of parameters per block. Consequently, the granularity of the partitioning determines the trade-off between computational cost and analytical resolution:

- **Coarse granularity** (e.g., treating an entire neuron or layer as a single block) involves larger parameter groups, leading to higher memory demand but faster overall computation. Under

this configuration, models with approximately up to 3B parameters can be handled efficiently on modern GPUs such as the NVIDIA H100 (80 GB).

- **Fine granularity** (e.g., defining blocks at the neuron-in-attention-head level) reduces per-block memory requirements substantially, enabling analysis of larger models, up to the 7–13B parameter range on comparable hardware.

In practice, the same principle generalizes across different hardware configurations. Devices with greater memory capacity or multi-GPU setups naturally extend the upper bound of model size, while smaller devices can still employ **MODEL SHAPLEY** effectively by adopting coarser partitioning. This hardware-agnostic flexibility allows practitioners to tailor the granularity–efficiency trade-off according to available resources, ensuring the method's applicability to a wide range of model scales and hardware environments.

# I   Broader Impacts

Our work leverages cooperative-game–theoretic Shapley values to develop **MODEL SHAPLEY**, an efficient, single-back-propagation estimator that assigns an importance score to every parameter in a neural network. By collapsing the naïve $2^M$-scale computation to one gradient pass, **MODEL SHAPLEY** produces interpretable saliency maps that drive targeted fine-tuning, pruning, and quantization, circumventing the accuracy degradation often seen with full-model fine-tuning. This efficiency is especially critical in the era of large models, where the sheer parameter count makes exhaustive attribution prohibitively expensive. In practice, **MODEL SHAPLEY** excels at large-model interpretability tasks (e.g., probing and logit-lens analysis), parameter-specific adaptation, and precision-aware quantization, providing a practical template for future responsible and efficient model development.

# J   Data Ethics Statement

To evaluate the efficacy of this work, we conducted experiments that only use publicly available datasets, namely, MMLU, GSM8K, CIFAR-100, and ImageNet in accordance with their usage terms and conditions, if any. We further declare that no personally identifiable information was used, and no human or animal subject was involved in this research.

