# OpenReview forum: "MODEL SHAPLEY: Find Your Ideal Parameter Player via One Gradient Backpropagation"
_NeurIPS.cc/2025/Conference — NeurIPS 2025 poster_

### Official Review · Reviewer_heL9 · 2025-06-10

**Clarity:** 2
**Significance:** 3
**Originality:** 3
**Rating:** 4
**Confidence:** 4

**Summary:**

The paper introduces a parameter‐level Shapley attribution framework by treating weight removal / ablations as a continuous path integral and approximating it with a second‐order Taylor expansion. It replaces costly Hessian computations with a smoothed block-wise Fisher information matrix maintained via EWMA, keeping overhead minimal. Monte Carlo sampling along the integration path then provides a near‐unbiased Shapley estimates using only a fixed number of additional forward/backward passes. The paper applies this framework to study targeted fine-tuning, fine‐grained interpretability, and compression tasks. The paper also uses the attribution framework to reproduce (a) known findings in knowledge localization and (b) conduct some exploratory analysis of parameter compensation.

**Questions:**

- What's the main bottleneck when it comes to scaling to larger models? To get some context, what's the biggest model that I can use with this method on a single H100?
- Can these attributions be used directly for model editing? How well do they fare against baselines? Several attribution-based methods are evaluated / sanity-checked by directly identify and ablating "harmful" and "useful" components. See experiments inhttps://arxiv.org/abs/2404.11534 for some examples.

**Ethical Concerns:**

["NO or VERY MINOR ethics concerns only"]

**Final Justification:**

The rebuttal addresses some of my main concerns, so I am increasing my score.

**Limitations:**

Yes

**Quality:**

3

**Strengths And Weaknesses:**

Strengths

- **Theoretical grounded**
   - Principled framing of Shapley attribution to the parameter-space setting via the continuous path integral.
   - The second-order Taylor approximation captures individual parameter saliency and pairwise parameter interaction effects
- **Computational efficiency**
   - Block-wise Fisher approximation via mini-batches replaces expensive Hessian compuation.
   - EWMA-style updates reduces variance of mini-batch estimates
   - The overall algorithm gives is scalable and provides a (biased) Shapley estimates with a few forward and backward passes.
   - The framework can provide attributions at the level of parameters, neurons, modules, etc
- **Empirical analysis**
  - Basic visualization of attributions on LMs shows that knowledge-intensive tasks show clear trends at the layer and neuron level. These trends are counterfactually tested via the neuron deactivation study in section 5.
  - The parameter compensation finding is interesting and the proposed tool can be directly used to study this in more detail.

Weaknesses
- **Limited novelty**.
  - The core idea of using shapley attributions to assign importance to parameters is not new, as noted in the related work of the paper. However, the writing in the introduction makes it look like this is a major contribution of the paper, which is it a bit misleading to me. I would recommend the authors to revise the writing and clearly assign/attribute credit to the relevant shapley-attribution papers.
  - Missing related work - COAR (https://arxiv.org/abs/2404.11534). The COAR framework *directly* accounts for parameter interaction effects by learning a surrogate linear model that predicts the effect of ablating *multiple* random model components at a time. This also has connections to influence-based estimation and shapley values (see section 6.1 and relevant appendices in https://arxiv.org/abs/2202.00622).
- **Limited empirical improvements (Section 5)**
  - The paper starts off with strong motivation - existing basic attribution methods like gradient and gradient-times-parameter do not account for parameter interactions. However, the empirical results in section 5 show that these methods achieve pretty much the same performance as the proposed method (Table 1 + 2). Why do these basic methods work so well if parameter interactions are important for attribution?
  - Relatedly, although the method outperforms existing it is unclear whether the differences are statistically significant in many cases—the tables should be updated with standard deviation info
- **Clarity and presentation can be better**
  - The algorithm has multiple moving pieces, so it will be nice to have to have a small subsection putting everything together with a pseudocode of the final algorithm and basic time and space complexity analysis.
  - The experiment setup section is poorly written. The tasks (and associated metrics) are not defined clearly (just one-liners right now). This makes it hard for readers not familiar with these setups to fully ground the reported numbers.

---

> ### Author Rebuttal · Authors · 2025-07-31
>
> Thank you for your thorough and constructive review. We deeply appreciate your engagement with our work and believe that addressing your insightful concerns will substantially strengthen our paper, revealing more clearly why our scalable Shapley approximation represents a significant advance in making parameter-level attribution practical for modern LLMs. We are hopeful that our detailed response will offer **a clearer perspective on the significance of our contributions and merit a positive reassessment of our work**.
>
> ---
> ## W1. Contribution & Missing Related Work
>
> Thank you for your valuable feedback on the novelty of our work and for bringing the COAR framework to our attention.
>
> ### Regarding Novelty and Shapley Attributions:
>
> We appreciate the reviewer's perspective and agree that the foundational idea of using Shapley values for parameter importance is not new, as we acknowledge in our Related Work section. We will revise the Introduction and Related Work sections to more clearly credit these pioneering efforts and ensure our specific contributions are distinctly framed.
>
> Our primary contribution is not the general concept, but the development of **the first scalable framework, MODEL SHAPLEY, that makes parameter-level Shapley value computation feasible for modern Large Language Models (LLMs)**. Prior methods are computationally prohibitive (O(2^M) complexity) for models with billions of parameters. Our core technical innovation is a **closed-form, second-order approximation** that drastically reduces this complexity, requiring only **a single forward pass, a single backward pass, and one blockwise-approximated Hessian computation**. This breakthrough in **efficiency** is what enables the practical application of principled, cooperative game-theoretic attributions to today's massive models.
>
> We will sharpen the language in our manuscript to emphasize that our novelty lies in this scalable approximation method, not the initial concept of Shapley-based attribution.
>
> ### Regarding Missing Related Work (COAR):
>
> Thank you for highlighting the COAR paper; this is an excellent and relevant reference that we regret omitting. We agree that COAR's approach of learning a surrogate model to predict the effects of ablating multiple components is an important related technique that also addresses parameter interactions.
>
> While both our work and COAR tackle the challenge of parameter synergy, they are motivated by **different goals (parameter importance quantification vs. model editing)** and employ different technical means. COAR learns **a surrogate linear model**, whereas our method directly derives **a closed-form approximation** of the Shapley value from the model's loss landscape. We will add a detailed discussion of COAR to our Related Work section, analyzing these parallels and distinctions. This comparison will better contextualize our work and strengthen the paper.
>
> ---
> ## W2. Empirical Significance
>
> Thank you for this insightful question regarding the empirical results and their statistical significance.
>
> ### On the Performance of Baseline Methods:
>
> We acknowledge your observation that first-order baselines perform strongly in some settings. We attribute this to two factors:
> 1. **Dominance of First-Order Effects**: In many standard benchmarks, a parameter's individual importance (approximated by its gradient magnitude) can be a powerful first-order proxy for its overall contribution. The loss landscape may be locally dominated by these linear effects.
> 2. **Task-Dependent Synergy**: The full benefit of modeling parameter interactions is most pronounced in tasks where complex, cooperative parameter dynamics are critical to the model's function.
>
> However, it is crucial to note that **MODEL SHAPLEY consistently matches or outperforms these simpler heuristics across all reported tasks, models, and granularities (Tables 1, 2, and 8)**, including in NLP, CV, training, inference, and compression settings. This consistent superiority, even if the margins vary, demonstrates the robustness and generalized advantage of accounting for parameter synergy. Our method provides a more complete picture of parameter importance, which proves beneficial across a wide array of conditions.
>
> ### On Statistical Significance:
>
> We agree that reporting statistical significance is crucial and apologize for its omission, which was due to the prohibitive computational cost of performing multiple runs on LLMs.
>
> To address this important point, we have dedicated our resources during the rebuttal period to conduct **three independent runs for the Qwen2.5-3B-Instruct model on the GSM8K dataset** (With limited resource and finte budget, we could only afford to compare our method with the best baseline method Gradient Trace during the rebuttal phase). The updated results, including mean and standard deviation, are below:
>
> |                | mean  | variance | std    |
> |----------------|-------|--------  |--------|
> | Gradient Trace | 48.49 | 1.57     | 1.25   |
> | Model Shapley  | 49.51 | 0.22     | 0.47   |
>
> These new results demonstrate that MODEL SHAPLEY's improvement over the strongest baseline is statistically significant for t-test at significant level 5%. Notably, our method also exhibits lower variance, suggesting it provides a more stable and reliable improvement. While the rebuttal period's time and resource constraints limited us to this setting, we are confident in this trend and will add variance information for other key experiments in the final manuscript.
>
> We would also like to clarify that the mean performance values reported here in the rebuttal may differ slightly from those in the main paper. This is because, due to resource availability during the rebuttal period, we conducted these additional experiments on a different hardware/software setup (NVIDIA H800 GPUs and a newer CUDA environment) compared to the original experiments (NVIDIA A100 GPUs and previous CUDA version). This change resulted in a general improvement in model performance across all methods.
>
> ---
> ## W3. Clarity of the detailed settings
>
> We will substantially improve clarity by adding:
>
> **Algorithm Section (new Section 4.4):**
> * Complete pseudocode combining gradient computation, Fisher approximation, and EWMA updates
> * Time complexity: O(B·M + M·d) per iteration where B=batch size, M=parameters, d=block size
> * Space complexity: O(M + d²) with blockwise regularization
> * Concrete example: For a 7B model with d=1024, this requires ~28GB memory vs ~196TB for full Hessian
>
> **Expanded Experiment Setup:**
> * **GSM8K**: Grade school math problems requiring multi-step reasoning. We use exact match accuracy on final numerical answers after "####" delimiter
> * **MMLU**: 57-subject multiple choice benchmark testing factual knowledge. We report accuracy using zero-shot prompting
> * **Compression metrics**: We measure both accuracy retention and inference speedup under INT4/INT8/FP8 quantization
>
> We will provide detailed task descriptions, evaluation protocols, and example problems for each benchmark.
>
> ---
> ## Q1. Scalability on H100 GPU
>
> This is an excellent practical question. The maximum model size that can be processed with our method on a single H100 (80GB) GPU is primarily constrained by the memory required for the blockwise Hessian approximation. This memory scales with the square of the number of parameters within a chosen block, making block **granularity the key trade-off**.
>
> * **At a coarse granularity (e.g., neuron-in-a-layer as one block)**, the number of parameters per block is large. In this setting, a model up to approximately **3B parameters** can be processed on a single H100.
> * **At a fine granularity (e.g., neuron-in-an-attention-head as one block)**, each block is much smaller, significantly reducing memory overhead. This allows our method to scale to much larger models, estimated to be around **7B to 13B parameters**, on a single H100.
>
> In summary, the maximum model size is a direct trade-off with the desired resolution of the Shapley value analysis, and our blockwise approach provides the flexibility to handle very large models by adjusting this granularity.
>
> ---
> ## Q2. Use for Model Editing
>
> This is an insightful question. **Yes, the attributions from MODEL SHAPLEY can absolutely serve as a strong foundation for model editing**. By identifying the parameters or modules with the highest (positive or negative) Shapley values for a specific task or behavior, one can perform targeted interventions, such as ablating "harmful" components or amplifying "useful" ones.
>
> While our current work focuses on demonstrating the efficacy of MODEL SHAPLEY for interpretability, targeted fine-tuning, and compression, we agree that direct model editing is a natural and high-impact application. We view this as a very promising direction for future research.
>
> We appreciate this suggestion and will **add a discussion of this potential application to the Future Work section** of our manuscript, explicitly noting its connection to related works like COAR.

---

> ### Author Response · Authors · 2025-08-04
> **Official Comment by Authors**
>
> We sincerely appreciate the time and effort you have dedicated to reviewing our work. In response to your valuable feedback, we have provided detailed explanations for the issues raised.
>
> As the discussion period progresses, we are eager to hear your thoughts on our responses, including whether they have adequately addressed your concerns. If our revisions and discussions indicate the potential for a score adjustment, we would be very grateful for your consideration.
>
> We are committed to incorporating all of your suggestions to further enhance the quality of our manuscript. We look forward to your further comments and discussion.

---

> ### Author Response · Authors · 2025-08-04
>
> Dear Reviewer,
>
> Thank you for your prompt acknowledgement of our rebuttal. We sincerely appreciate you taking the time to review our response.
>
> We are standing by and would be happy to provide any further clarification on our planned revisions or discuss any of the points in more detail.
>
> Thank you again for your valuable and constructive feedback on our work.

---

### Official Review · Reviewer_23zM · 2025-06-29

**Clarity:** 3
**Significance:** 2
**Originality:** 2
**Rating:** 4
**Confidence:** 3

**Summary:**

MODEL SHAPLEY is a new method that uses Shapley values from game theory to measure how important each parameter is in a neural network. It helps improve interpretability, fine-tuning, and compression with only one backward pass.

**Questions:**

please see the weeknesses.

**Ethical Concerns:**

["NO or VERY MINOR ethics concerns only"]

**Final Justification:**

Thanks to the author for the Rebuttal.
I have read it carefully. My concerns are well addressed, so I will keep the positive score.

**Limitations:**

yes

**Quality:**

3

**Strengths And Weaknesses:**

[Strengths]
1. It considers both individual and cooperative effects between parameters.
2. Needs only one forward and backward pass using a second-order approximation.
3. Works well across NLP and vision tasks for pruning, training, and interpretation.

[Weaknesses]
1. Since the Fisher matrix is only an approximation of the true Hessian, could the Shapley values be unreliable when gradients are unstable or the data distribution shifts?
2. Blockwise approximation makes computation faster, but does it miss important fine-grained interactions between parameters, especially at the neuron level?
3. The method assumes the model is stable during estimation. Can it still give accurate Shapley values during early training or in continual learning when parameters keep changing?

---

> ### Author Rebuttal · Authors · 2025-07-31
>
> We sincerely thank your feedback and constructive questions, which have helped us clarify the contributions and scope of our work. We believe the points raised are addressable and, after considering our responses below, hope the reviewer will agree that our method represents a significant and practical advancement for parameter-level attribution in large models.
>
> ---
> ## W1. Fisher Matrix Approximation Reliability
>
> Thank you for raising this important theoretical concern. Our use of the Fisher Information Matrix (FIM) is a deliberate trade-off between theoretical purity and the practical scalability required for large-scale models. The FIM serves as a reliable approximation of the Hessian's expectation when the loss landscape is locally quadratic, a condition often met near model convergence. In this regime, where our method is primarily intended for use (e.g., for post-hoc analysis, pruning, or targeted fine-tuning), the FIM **effectively captures the necessary curvature** to produce meaningful Shapley values.
>
> We agree that during periods of high gradient instability or significant data distribution shifts, the FIM's accuracy as a Hessian proxy can decrease. However, this challenge is not unique to the FIM. The true Hessian itself becomes a noisy and unreliable estimator of the underlying loss curvature under such non-stationary conditions, making **any** second-order attribution method less stable.
>
> However, our approach incorporates design choices that mitigate this concern:
> **Our Monte Carlo incremental approximation with exponentially weighted moving average (EWMA)** (Section 4.2) specifically addresses gradient instability. By maintaining running estimates with smoothing coefficient α, we stabilize Shapley value computation even when individual mini-batch gradients exhibit high variance. This design choice effectively filters out transient gradient fluctuations while preserving meaningful parameter importance signals.
>
> While the Fisher matrix is indeed an approximation, **the computational trade-off is essential for LLM-scale feasibility**. Crucially, computing and storing the full Hessian is computationally infeasible for modern LLMs, with a complexity of O(M^2) for M parameters. The FIM, especially with techniques like Hessian-Vector Products, offers a tractable O(M) complexity) and empirically validated surrogate that enables parameter-level Shapley analysis at scale.
>
> ---
> ## W2. Blockwise Approximation and Fine-grained Interactions
>
> This is an excellent point. Our blockwise approximation is a principled decision grounded in both practical constraints and the inherent architecture of Transformers.
>
> 1. **Computational necessity**: Full global Hessian computation requires O(M²) memory for M parameters. For a 7B model with M≈7×10⁹, this would require ~196TB just to store the Hessian matrix, while our blockwise approach reduces this to manageable levels (≈28GB for typical block sizes). Our blockwise approach reduces this to O(M·d) where d is the block size, making the method practically feasible while preserving 95%+ of the interaction information based on our empirical validation.Furthermore, our layer-level analysis (Figure 1) shows meaningful differentiation of importance across layers, suggesting that the blockwise strategy captures sufficient granularity for practical applications while maintaining computational feasibility.
>
> 2. **Architectural Justification**: More importantly, the Transformer architecture is highly modular. Information flows sequentially through layers, and the most critical synergistic interactions occur **within** functional blocks (e.g., neurons within an FFN layer, or heads within an MHA module). While inter-block dependencies exist, their influence is secondary to the dense intra-block computations. Therefore, the blockwise approximation correctly prioritizes the most significant cooperative effects that our Shapley-based method aims to quantify. Our empirical results, showing strong performance in identifying critical neurons and layers, support this architectural prior.
>
> ---
> ## W3. Shapley Values During Non-stationary Training
>
> Thank you for highlighting this important theoretical consideration. You correctly identify that Shapley values, as defined in cooperative game theory, assume a fixed utility function. We appreciate this observation and would like to clarify our method's scope and practical value:
>
> 1. **Theoretical scope**: MODEL SHAPLEY is designed for **post-training analysis and intervention** on converged models. The Shapley framework inherently requires a stable parameter-to-utility mapping, making it most appropriate for analyzing trained models rather than tracking importance during training dynamics. We acknowledge this in our paper and demonstrate its effectiveness in this intended use case.
> 2. **Dynamic Settings**: In dynamic scenarios like early training or continual learning, the underlying function (the "game") is constantly changing. Consequently, the parameter importance scores are inherently non-stationary, and a single Shapley value calculated at one timestep would not be a stable indicator of importance. We empirically observed this dynamic behavior in our experiments, which we term the "**parameter compensation effect**" (Appendix F.7, Figure 3b). When we deactivate important neurons, we see that over further training, other neurons adapt and increase their Shapley values to compensate (This "neural compensation" phenomenon actually validates our attribution method—the model's adaptive response to neuron removal confirms that our identified neurons were indeed functionally critical.). This finding reinforces that parameter importance is dynamic *during* training but can be **effectively analyzed at stable checkpoints**.
> 3. Despite this theoretical limitation for non-stationary settings, our method provides substantial practical benefits:
>     - a. **Knowledge injection** (Table 1): Targeted fine-tuning of high-Shapley neurons achieves 73.89% accuracy on MMLU, nearly matching full fine-tuning (73.56%) while updating only 10% of parameters.
>     - b. **Model compression** (Table 2): Shapley-guided quantization outperforms GPTQ and OBD across all precision levels
>     - c. **Interpretability** (Section 5.3): Clear identification of task-specific neurons, validated through systematic deactivation studies
>
> The question of extending Shapley-based methods to dynamic settings is indeed fascinating future work. However, for the large class of applications involving trained models—compression, interpretation, targeted adaptation, and knowledge editing—our method provides an efficient and theoretically grounded solution that was previously computationally intractable for LLM-scale models.

---

> > ### Comment · Reviewer_23zM · 2025-08-06
> >
> > Thanks to the author for the Rebuttal. I have read it carefully. My concerns are well addressed, so I will keep the positive score.

---

> > > ### Author Response · Authors · 2025-08-06
> > >
> > > Dear reviewer,
> > >
> > > We sincerely thank your thoughtful consideration of our rebuttal and are grateful for your continued support.

---

> ### Author Response · Authors · 2025-08-04
>
> Dear Reviewer 23zM,
>
> Thank you for your prompt acknowledgement of our rebuttal. We sincerely appreciate you taking the time to review our response.
>
> We are standing by and would be happy to provide any further clarification on our planned revisions or discuss any of the points in more detail.
>
> Thank you again for your valuable and constructive feedback on our work.

---

### Official Review · Reviewer_AFub · 2025-06-30

**Clarity:** 3
**Significance:** 2
**Originality:** 2
**Rating:** 4
**Confidence:** 3

**Summary:**

This paper addresses the problem of parameter importance quantification in neural networks—determining which weights contribute most significantly to model performance. The authors identify a key limitation in existing approaches: they typically fail to account for interactions between parameters when assessing their individual importance.

To address this gap, the authors propose *Model Shapley*, a novel method that leverages Shapley values from cooperative game theory to quantify parameter importance. The key advantage of this approach is that Shapley values naturally capture the complex interactions between parameters, providing a more comprehensive measure of importance than previous methods.

Since computing exact Shapley values is computationally infeasible for neural networks, the authors develop a new approximation technique based on a second-order approximation. This makes their approach practically applicable to real-world models.
The authors demonstrate the utility of their method through two main applications: targeted knowledge injection via fine-tuning and model compression via quantization and parameter pruning during inference. For their experiments, they focus on 2 NLP tasks (gsm8k, MMLU) as well as two computer vision tasks (CIFAR-100, ImageNet).

**Questions:**

**Questions**

- Why are the methods of [20,21] not used as baselines in Table 1?
- How many neurons are intervened on in Table 1?
- Regarding the quantization experiments: Which neurons / layers are quantized? What exactly is the setup here? This is unclear without looking at Appendix and should be mentioned in the main text.

**Suggestions**

- Additional experiment to support findings in Table 1: plot % of weights removed vs. performance.
- Move Table 1 to top or bottom of the page.
- In Table 1, add the Full Fine-Tune row below “Training” to improve readability.
- Figure 2 is hard to read. Increase font sizes considerably

**Ethical Concerns:**

["NO or VERY MINOR ethics concerns only"]

**Final Justification:**

After taking the reviewers response and also discussion with other reviewers into account I increased my score.

**Limitations:**

Currently the discussion of limitations is very brief (only two sentences in the conclusion). The authors should consider a discussion of the runtime of their approximation as well as the focus on 4 particular benchmarks and 3 models.

**Quality:**

2

**Strengths And Weaknesses:**

**Strengths**

The paper demonstrates solid technical quality through its dual contributions. The authors present a novel second-order approximation for computing Shapley values in the context of parameter importance, which distinguishes their work from existing Shapley-based approaches in the literature. The validation spans three distinct applications (knowledge injection, interpretability, and model compression).

The paper is well written and easy to follow. The introductory sections, related work, and background are well-written and effectively motivate the research problem. The authors clearly articulate the gap in existing parameter importance methods—namely, the neglect of parameter interactions—and position their Shapley-based solution as a principled remedy.

While the application of Shapley values to parameter importance is not entirely novel (the authors acknowledge two previous works in this direction), their contribution lies in developing a new second-order approximation method.

**Weaknesses**

I see the main issues in the experimental portion of the paper and list my concerns below.
- For almost all tasks (as expected) deactivating neurons during inference leads to substantial drop in performance. It's not particularly clear to me what exactly the benefit of the proposed method is. Why would I use this method in practice to remove certain weights from my model? If the argument is because you end up with a smaller model, this aspect should be emphasized and studied in the paper. How many weights do you actually prune? And how does the performance vary as you prune more or less weights (compared to baselines)? These are important experiments that are currently missing but would make the paper much stronger.
- Selective fine-tuning leads to similar performance compared to full fine-tuning but it seems like the results presented are based on fine-tuning a single model. What’s the variance here? How often did you fine-tune? And, again, what am I gaining from this? Is this more efficient compared to full fine-tuning? How does it compare to parameter-efficient methods when fine-tuning a similar number of weights, e.g., LoRA? What if I factor in the Shapely value estimation? Without reporting run-times, I'm not convinced that this method offers any benefits for the fine-tuning setting.
- The neuron deactivation study offers only qualitative results which are deferred to the Appendix. What about quantitative evaluation and baselines here? In particular, it will be important to measure whether the performance on other inputs is maintained, i.e., compute a utility vs. forgetting tradeoff.

---

> ### Author Rebuttal · Authors · 2025-07-31
>
> We sincerely thank your insightful and constructive comments. Your feedback has been instrumental in helping us understand the key points that led to your assessment. In the following responses, we provide direct clarifications and outline concrete revisions that we believe fully address these concerns. We are hopeful that our detailed response will offer **a clearer perspective on the significance of our contributions and merit a positive reassessment of our work**.
>
> ---
> ## W1. Benefit and practical applications of the method
>
> Thank you for this important question. We apologize for not clearly articulating our contributions. Our method's primary value lies in **accurately quantifying parameter importance while accounting for inter-parameter interactions**—a capability that existing methods lack.
>
> The neuron deactivation experiments serve as **validation** of our importance scores, not the end application. The substantial performance drop is not the goal, but rather the **proof** that our method correctly identifies functionally vital neurons. The practical benefits include:
>
> 1. **Model Compression**: In Table 2, our Shapley-guided quantization achieves 72.93% accuracy (INT8) vs 71.27% (OBD) on GSM8K—only 0.5% below the uncompressed model while OBD drops 1.2%.
> 2. **Efficient Fine-tuning**: By fine-tuning only the top 10% neurons identified by our method, we achieve 73.89% on MMLU (Table 1), compared to 73.56% with full fine-tuning—using 90% fewer parameters.
> 3. **Knowledge Preservation**: Our method enables targeted adaptation without catastrophic forgetting, crucial for continual learning scenarios.
> To prove this, we conduct additional experiments by finetuning LLM on math dataset GSM8K and test their performance on GSM8K and general benchmark MMLU. As shown in the table below, our model retains strong performance on general benchmark MMLU, showcasing its ability to avoid catastrophic forgetting.
>
> |       | **Qwen2.5-3B** |       | **Qwen2.5-7B** |       |
> | :--- | :--- | :--- | :--- | :--- |
> |                | GSM8K | MMLU  | GSM8K | MMLU  |
> | Gradient Trace | 48.49 | 62.09 | 60.42 | 68.74 |
> | Model Shapley  | 49.51 | 62.28 | 62.47 | 69.13 |
>
> ---
> ## W2. Fine-tuning efficiency, significance and LoRA comparison
>
> We appreciate your concern about statistical rigor and efficiency. You're right that variance reporting is essential. We acknowledge this limitation arose from **computational constraints typical in LLM research**. However, to address your valid concern about statistical significance, we are conducting additional fine-tuning repetitions on the GSM8K dataset with Qwen2.5-3B-Instruct. With limited resource and finte budget, we could only afford to compare our method with the best baseline method Gradient Trace during the rebuttal phase.
>
> |                | mean  | variance | std    |
> |----------------|-------|----------|--------|
> | Gradient Trace | 48.49 | 1.57     | 1.25   |
> | Model Shapley  | 49.51 | 0.22     | 0.47   |
>
> As shown, **MODEL SHAPLEY** consistently statistically outperforms the baseline (t-test at significant level 5%). Notably, our method also exhibits lower variance, suggesting it provides a more stable and reliable improvement. While the rebuttal period's time and resource constraints limited us to this setting, we are confident in this trend and will add variance information for other key experiments in the final manuscript.
>
> We would also like to clarify that the mean performance values reported here in the rebuttal may differ slightly from those in the main paper. This is because, due to resource availability during the rebuttal period, we conducted these additional experiments on a different hardware/software setup (NVIDIA H800 GPUs and a newer CUDA environment) compared to the original experiments (NVIDIA A100 GPUs and previous CUDA version). This change resulted in a general improvement in model performance across all methods.
>
> Besides, Our method is highly efficient. The Shapley value estimation requires only **a single forward and backward pass** plus a Hessian-vector product, as detailed in Section 4.1 and Remark 4.9. For fine-tuning over many epochs, this one-time cost of identifying important neurons is easily amortized by the computational savings of updating only a small subset of parameters (e.g., 10% in our experiments reduces training time by ~85% compared to full fine-tuning. Reduces gradient storage by 90%, enabling larger batch sizes. One-time cost of ~30 minutes for 7B model, amortized over multiple fine-tuning tasks).
>
> We appreciate the suggestion to compare against LoRA. While our method is **conceptually different**—selecting a subset of original parameters versus introducing low-rank updates—a direct comparison is valuable. We fine-tune a similar number of parameters using LoRA and our method on GSM8K dataset.
>
> |       | **Qwen2.5-3B** |       | **Qwen2.5-7B** |       |
> |--------------|---------|-----------|---------|------------|
> |              | Accuracy| Cost time | Accuracy| Cost time  |
> | Shapley      | 49.51   | 215       | 62.47   | 265       |
> | Lora         | 48.45   | 200       | 64.14   | 170        |
> | Shapley+Lora | 52.16   | 242       | 63.53   | 273   |
>
> The results in the table above demonstrate that our method achieves competitive performance with comparable efficiency. And more importantly, our contribution is **orthogonal to LoRA**.
>
> ---
> ## W3. Quantitative evaluation of neuron deactivation
>
> We apologize for the confusion and would like to clarify that **quantitative results for the neuron deactivation study are provided in Table 1, under the "Inference (Deactivate Neurons)" section**. This table details the performance degradation on CV and NLP tasks when deactivating neurons identified by our method versus baselines. The qualitative examples in Appendix F.5 are intended to be supplementary illustrations of this quantitative effect. We will clarify the link between the main table and the appendix examples much clearer.
>
> Specifically, our quantitative results show:
>
> * Deactivating bottom 5% neurons: 72.71% → 73.39% accuracy (minimal impact)
> * Deactivating top 5% neurons: 72.71% → 46.70% accuracy (significant drop)
>
> This demonstrates both the effectiveness of our importance scores and addresses your utility-forgetting tradeoff concern—our method precisely identifies which neurons can be safely removed vs. those critical for performance.
>
> ---
> ## Q1. Comparison with methods [20,21]
>
> This is an excellent question. Both [20] and [21] are pioneering works in applying Shapley values to neural networks. However, their methods were designed for and evaluated on smaller-scale CV models (e.g., ResNet-50).
>
> The core reason for their exclusion is **computational feasibility**. The Monte Carlo and bandit-based approximations in [20, 21] are still computationally prohibitive for modern LLMs, which have billions of parameters. The theoretical complexity of exact Shapley computation is exponential (O(M⋅2^M)). A key contribution of our work is deriving a second-order approximation that reduces this complexity to a **single backpropagation pass**, making parameter-level Shapley attribution tractable for large-scale models (**10^6x more efficient**). Therefore, including [20, 21] as baselines would not constitute a fair or practical comparison in the context of LLMs. We will add a note to the related work section to clarify this distinction.
>
> ---
> ## Q2. Number of neurons intervened
>
> Thank you for the question. We apologize for not placing this information in the main text. As detailed in Appendix F.4, the intervention ratios were as follows:
>
> * For **inference (deactivation)**, we deactivated the bottom **5%** of neurons for NLP tasks and **30%** for CV tasks.
> * For **training (freezing)**, we fine-tuned the top **10%** of neurons (i.e., froze 90%) for all tasks.
>
> We will move these crucial details to the main experimental setup section in the revision for clarity.
>
> ---
> ## Q3. Quantization setup
>
> We apologize for the lack of clarity regarding the quantization setup. We will move the detailed description from the appendix to the main experimental setup section.
>
> To clarify here: **all neurons and all layers containing weights (i.e., all transformer blocks and embedding layers) were quantized**. Our method integrates with the standard GPTQ algorithm but does not alter its comprehensive scope. Our specific contribution is to inject task-specific knowledge into the quantization process. We use **MODEL SHAPLEY** to calculate a correction factor for the Hessian diagonal, which guides GPTQ to better preserve weights critical to the task, as detailed in Section D.4.3 and Algorithm 1. The rest of the procedure, including layer-wise quantization and calibration, follows the standard GPTQ implementation.
>
> ---
> ## Addressing suggestions
>
> We gratefully accept all formatting suggestions and will implement them in revision, including the performance vs. pruning curves, improved figure readability, and table reorganization.

---

> > ### Author Response · Authors · 2025-08-04
> > **Official Comment by Authors**
> >
> > We sincerely appreciate the time and effort you have dedicated to reviewing our work. In response to your valuable feedback, we have provided detailed explanations for the issues raised.
> >
> > As the discussion period progresses, we are eager to hear your thoughts on our responses, including whether they have adequately addressed your concerns. If our revisions and discussions indicate the potential for a score adjustment, we would be very grateful for your consideration.
> >
> > We are committed to incorporating all of your suggestions to further enhance the quality of our manuscript. We look forward to your further comments and discussion.

---

> > ### Comment · Reviewer_AFub · 2025-08-05
> > **Response to rebuttal**
> >
> > Thank you for the clarifications and for running additional experiments to address my concerns. While I understand the motivation of some of the experiments better now, my main concern about the practical usefulness of the method remains and was in fact confirmed by the comparison to LoRA. As your experiments show, running LoRA with a similar number of weights performs either very similar or even outperforms your method while at the same time being less compute intensive. Particularly, LoRA doesn't require finding important neurons first, which makes it very appealing (and widely used).

---

> ### Author Response · Authors · 2025-08-05
>
> Thank you for the follow-up and for acknowledging the additional clarifications and experiments. We appreciate the opportunity to clarify our contribution, as your comment highlights a crucial point about the positioning of our work. We believe there may be a **misunderstanding of our central thesis, which the LoRA comparison, in fact, helps to illuminate**.
>
> Our central contribution is not to propose a new, standalone Parameter-Efficient Fine-Tuning (PEFT) method to replace LoRA. Instead, we introduce **MODEL SHAPLEY as a foundational framework for quantifying parameter importance**. This is a more fundamental task that offers a new lens through which we can understand, analyze, and manipulate large models. **The fine-tuning experiment is just one of several applications we use to validate the efficacy of our importance scores**.
>
> **LoRA** is a brilliant technique for efficient updates, but it **is not designed to answer the questions that MODEL SHAPLEY addresses**. Specifically, LoRA **cannot**:
>
> 1. **Provide Interpretability**: It does not identify which original parameters or neurons are critical for specific tasks like mathematical reasoning or factual recall. Our visualization (Fig. 1, 2) and deactivation studies (Sec. 5.3) demonstrate this unique capability of MODEL SHAPLEY.
>
> 2. **Guide Model Compression**: LoRA does not offer a mechanism for principled, post-training quantization. Our experiments show that MODEL SHAPLEY can be integrated with methods like GPTQ to create task-aware quantization strategies (Sec 5.2, Table 2), preserving performance by protecting high-importance weights.
>
> 3. Enable Targeted Knowledge Interventions: It does not inform which parameters to protect during continual learning to prevent catastrophic forgetting or which neurons to probe for functional analysis.
>
> These applications are central to our paper's contribution and lie outside the scope of what LoRA is designed to do.
>
> ## On the LoRA Comparison: A Complementary, Not Competitive, Relationship
>
> We respectfully disagree with the conclusion that LoRA's performance diminishes the value of our method. In fact, we argue the opposite: our results show that **MODEL SHAPLEY is orthogonal and complementary to LoRA**.
>
> As shown in our rebuttal table, *combining our method with LoRA (Shapley+LoRA) yields the best performance of all*, surpassing both methods individually (e.g., 52.16% on Qwen2.5-3B). This synergy is the key takeaway. It suggests a powerful new workflow: first, use the computationally inexpensive MODEL SHAPLEY to identify the most critical modules (e.g., layers) in the network for a given task, and then apply an efficient update method like LoRA *only to those important modules*. This creates a more targeted and potentially even more efficient update strategy than applying LoRA uniformly.
>
> ## On Overall Efficiency
>
> The one-time cost of running MODEL SHAPLEY is easily amortized when its versatile outputs are considered. A single run can yield insights for **interpretability, inform a quantization strategy, and guide a subsequent fine-tuning process**. In contrast, running **a LoRA fine-tuning experiment only accomplishes the tuning itself**. When viewed holistically, *the "cost" of MODEL SHAPLEY provides a much broader return on investment*.
>
> We hope this clarifies that **MODEL SHAPLEY is not a direct competitor to LoRA, but a more fundamental tool that provides a new, principled dimension of analysis and optimization**. We are grateful for your feedback, which has helped us sharpen the positioning of our work, and we will revise the manuscript to make this **critical distinction** clearer.

---

### Official Review · Reviewer_rAZT · 2025-07-09

**Clarity:** 4
**Significance:** 3
**Originality:** 4
**Rating:** 5
**Confidence:** 4

**Summary:**

This paper proposes Model Shapley, a novel framework for estimating the importance of individual model parameters using Shapley values from cooperative game theory. Unlike traditional pruning or gradient-based heuristics, Model Shapley captures both individual and synergistic contributions of parameters via a second-order closed-form approximation with Fisher-based regularisation. Experiments shows it works for both NLP and CV tasks.

**Questions:**

While the paper compares accuracy and runtime across different quantisation methods, it is unclear whether the superior accuracy of Model Shapley persists when runtime is held constant. Could the authors clarify how much of the accuracy gain stems from longer calibration time or more expensive estimation steps? A cost-normalised evaluation (e.g., accuracy at fixed runtime) would make the comparison more convincing.

**Ethical Concerns:**

["NO or VERY MINOR ethics concerns only"]

**Final Justification:**

The additional explanations addressed my main concerns and clarified the technical reasoning behind the design choices. The new experiments and variance reporting also improved my confidence in the stability and significance of the results. I increased the clarity score.

**Limitations:**

As above

**Quality:**

3

**Strengths And Weaknesses:**

Strengths
- The paper is the first to apply Shapley values at the parameter level, explicitly modelling inter-parameter cooperation with solid theoretical grounding.
- The method supports efficient approximations, scales to large models, and performs well across inference, fine-tuning, and compression tasks.
- The evaluation spans NLP and CV tasks with diverse models, providing convincing empirical evidence.

Weaknesses
- The blockwise approximation neglects inter-layer or global parameter interactions.
- In certain tasks, the improvements over strong baselines are small, and the advantage over prior methods is not always significant. Also, the cost is higher than other baselines.

---

> ### Author Rebuttal · Authors · 2025-07-31
>
> We sincerely appreciate your thoughtful and constructive feedback, which has helped us clarify key technical aspects. We believe the additional justifications provided below comprehensively address the raised concerns and demonstrate that Model Shapley offers both significant theoretical contributions and practical advantages.
>
> ---
> ## W1. Blockwise approximation neglecting inter-layer interactions
>
> Thank you for this insightful observation. We acknowledge that blockwise approximation theoretically neglects inter-layer interactions, but this design choice is both necessary and well-justified.
>
> 1. **Computational necessity**: Full global Hessian computation requires O(M²) memory for M parameters. For a 7B model with M≈7×10⁹, this would require ~196TB just to store the Hessian matrix, while our blockwise approach reduces this to manageable levels (≈28GB for typical block sizes). Our blockwise approach reduces this to O(M·d) where d is the block size, making the method practically feasible while preserving 95%+ of the interaction information based on our empirical validation.Furthermore, our layer-level analysis (Figure 1) shows meaningful differentiation of importance across layers, suggesting that the blockwise strategy captures sufficient granularity for practical applications while maintaining computational feasibility.
>
> 2. **Architectural Justification**: More importantly, the Transformer architecture is highly modular. Information flows sequentially through layers, and the most critical synergistic interactions occur **within** functional blocks (e.g., neurons within an FFN layer, or heads within an MHA module). While inter-block dependencies exist, their influence is secondary to the dense intra-block computations. Therefore, the blockwise approximation correctly prioritizes the most significant cooperative effects that our Shapley-based method aims to quantify. Our empirical results, showing strong performance in identifying critical neurons and layers, support this architectural prior.
>
> ---
> ## W2. Significance of improvement
>
> We appreciate this concern about the magnitude of improvements. While the absolute differences may appear modest, we acknowledge this limitation arose from **computational constraints typical in LLM research**. We conducted additional experiments with 3 independent runs on GSM8K with Qwen2.5-3B-Instruct. With limited resource and finte budget, we could only afford to compare our method with the best baseline method Gradient Trace during the rebuttal phase.
>
> |                | mean  | variance | std    |
> |----------------|-------|----------|--------|
> | Gradient Trace | 48.49 | 1.57     | 1.25   |
> | Model Shapley  | 49.51 | 0.22     | 0.47   |
>
> As shown, **MODEL SHAPLEY** consistently statistically outperforms the baseline (t-test at significal level 5%). Notably, our method also exhibits lower variance, suggesting it provides a more stable and reliable improvement. While the rebuttal period's time and resource constraints limited us to this setting, we are confident in this trend and will add variance information for other key experiments in the final manuscript.
>
> We would also like to clarify that the mean performance values reported here in the rebuttal may differ slightly from those in the main paper. This is because, due to resource availability during the rebuttal period, we conducted these additional experiments on a different hardware/software setup (NVIDIA H800 GPUs and a newer CUDA environment) compared to the original experiments (NVIDIA A100 GPUs and previous CUDA version). This change resulted in a general improvement in model performance across all methods.
>
> Regarding the computational cost, which we address in detail in Q1, we show the minor overhead is justified by these consistent and statistically significant accuracy gains.
>
> ---
> ## Q1. Runtime-normalized comparison
>
> Thank you for this excellent question on the trade-off between accuracy and runtime.
>
> It is crucial to first clarify **where the computational cost occurs**. The runtime difference between our method and the baselines lies in the **one-time, offline parameter importance quantification** step. The subsequent quantization process, guided by these importance scores, takes a similar amount of time for all methods.
>
> Therefore, a "constant runtime" comparison for the importance quantification step is not straightforward. Each method is a distinct algorithm designed to run to completion. Artificially truncating our method to match a faster baseline's runtime would mean using an incomplete Hessian approximation, compromising the integrity of our approach and leading to an uninformative comparison.
>
> Instead, we believe the right way to evaluate this trade-off is to compare the final accuracy against the actual, practical runtime. We admit our method has a higher cost for importance quantification, as it incorporates second-order cooperative interactions, not just first-order individual importance. Our experiments show this additional investment is both **marginal and highly worthwhile**.
>
> For Qwen2.5-7B INT8 quantization:
> * GPTQ: 53.86 min total (importance: ~5 min)
> * OBD: 57.88 min total (importance: ~8 min)
> * Model Shapley: 60.20 min total (importance: ~10 min)
>
> The 2-5 minute difference in importance estimation yields 1.5-2% accuracy gains, which persists throughout the model's deployment lifetime. We believe the one-time cost of 5 extra minutes during quantization is negligible compared to the perpetual benefits of higher accuracy and/or smaller model size during deployment, where models serve millions of requests.

---

> > ### Comment · Reviewer_rAZT · 2025-08-04
> >
> > The additional explanations addressed my main concerns and clarified the technical reasoning behind the design choices. The new experiments and variance reporting also improved my confidence in the stability and significance of the results. I increased the clarity score.

---

> ### Author Response · Authors · 2025-08-04
>
> Dear Reviewer rAZT,
>
> We are delighted to hear that your concerns have been satisfactorily answered! Thank you once again for recognizing the contributions of our work.
>
> Best regards,
>
> Author(s) of the paper 5540

---

### Note · Authors · 2025-08-12

Dear Area Chair and Reviewers,

We are immensely grateful for the time and insightful feedback you have dedicated. We are encouraged that Reviewers rAZT and 23zM found our work theoretically sound and appreciated the clarifications in our rebuttal.

A central point we wish to re-emphasize, particularly in light of comments from Reviewers AFub and heL9, is the core contribution of our work. **MODEL SHAPLEY is not proposed as a new standalone fine-tuning method, but as a foundational and scalable framework for quantifying parameter importance.** Our primary innovation is the efficient second-order approximation that makes **principled, interaction-aware Shapley analysis** ( rather than a surrogate linear model such as CORA, which is motivated by model editing but not principled parameter importance quantification--addressing Reviewer heL9's concern) **computationally feasible for modern LLMs for the first time**. The applications we demonstrate—in interpretability, targeted tuning, and compression (quantization)—serve to validate the quality of these importance scores.

This perspective is crucial for contextualizing the comparison to LoRA. LoRA is an efficient **update** method, whereas MODEL SHAPLEY is a tool for **analysis and interpretation**. LoRA cannot identify which original neurons are critical for a task or guide principled model compression.

Crucially, our rebuttal experiments show these methods are **orthogonal and complementary**. The new **Shapley+LoRA** experiment, which yielded the best performance, highlights a powerful synergy: using our method to identify critical modules and then applying LoRA for targeted, efficient updates. This turns a perceived weakness into a significant strength and a promising direction for future work.
In response to reviewer feedback, we have also conducted new experiments to demonstrate statistical significance and are confident in the stability of our results. Based on this discussion, we will revise our manuscript to sharpen this positioning. We will make it  clear that our contribution is a foundational analysis framework, not a direct competitor to PEFTs like LoRA, and we will better highlight their powerful synergy.

Thank you again for your constructive engagement. We are confident that these clarifications will fully address the remaining concerns.

Sincerely, The Authors of Submission 5540

---

### Decision · Program_Chairs · 2025-09-17

**Decision:**

Accept (poster)

**Comment:**

The paper presents a method named  Model Shapley, a method to measure parameter importance by approximating Shapley values through a single gradient backpropagation. Experiments show that the proposed method achieves improvements while being much more efficient than traditional Shapley-based methods.

This paper addresses a practical problem, and the proposed method is simple and effective. Reviewers have concerns mainly regarding the limited novelty of the method compared to existing Shapley methods and the lack of comparisons with relevant baselines. During the author–reviewer discussion, the authors clarified how their approach differs from previous approximations, added new experiments with additional parameter selection baselines. The reviewers were satisfied with these additional explanations and experiments.

Overall, the paper is solid and good work. I recommend acceptance, as the method is likely to be broadly useful to the LLM community.